# Mutant *RIG-I* enhances cancer-related inflammation through activation of circRIG-I signaling

Jia Song[1,2,3,6], Wei Zhao[4,6], Xin Zhang[2], Wenyu Tian[2], Xuyang Zhao [2], Liang Ma[4], Yongtong Cao[4], Yuxin Yin [2,7] ✉, Xuehui Zhang [3,5,7] ✉, Xuliang Deng [1,5,7] ✉ & Dan Lu [2,7] ✉

RIG-I/DDX58 plays a key role in host innate immunity. However, its therapeutic potential for inflammation-related cancers remains to be explored. Here we identify frameshift germline mutations of *RIG-I* occurring in patients with colon cancer. Accordingly, *Rig-i^{fs/fs}* mice bearing a frameshift mutant *Rig-i* exhibit increased susceptibility to colitis-related colon cancer as well as enhanced inflammatory response to chemical, virus or bacteria. In addition to interruption of *Rig-i* mRNA translation, the *Rig-i* mutation changes the secondary structure of *Rig-i* pre-mRNA and impairs its association with DHX9, consequently inducing a circular RNA generation from *Rig-i* transcript, thereby, designated as circRIG-I. CircRIG-I is frequently upregulated in colon cancers and its upregulation predicts poor outcome of colon cancer. Mechanistically, circRIG-I interacts with DDX3X, which in turn stimulates MAVS/TRAF5/TBK1 signaling cascade, eventually activating IRF3-mediated type I IFN transcription and aggravating inflammatory damage. Reciprocally, all-trans retinoic acid acts as a DHX9 agonist, ameliorates immunopathology through suppression of circRIG-I biogenesis. Collectively, our results provide insight into mutant *RIG-I* action and propose a potential strategy for the treatment of colon cancer.

Unlike the pathogen-specific immune response, the nonspecific inflammatory lesions contribute to the malignant transformation of normal cells[1]. Despite the mechanism of inflammation-induced tumorigenesis has not been fully elucidated, prolonged inflammation has been verified to play a stimulatory role on DNA mutations and chromatin instability[2]. In addition to tumor initiation, the inflammatory condition also promotes cancer cell invasion and distal metastasis, and even increases cancer cell resistance to immunotherapy[3]. Emerging clinical studies show that chronic inflammation predisposes individuals to various types of cancer, including colon cancer, prostate cancer and liver cancer[4]. Therefore, elucidation of mechanism by which cancer-related inflammation is

---

[1]Department of Geriatric Dentistry, Peking University School and Hospital of Stomatology, Beijing 100081, P.R. China. [2]Institute of Systems Biomedicine, School of Basic Medical Sciences, NHC Key Laboratory of Medical Immunology, Beijing Key Laboratory of Tumor Systems Biology, Peking University Health Science Center, Beijing 100191, P.R. China. [3]National Engineering Research Center of Oral Biomaterials and Digital Medical Devices, NMPA Key Laboratory for Dental Materials, Beijing Laboratory of Biomedical Materials & Beijing Key Laboratory of Digital Stomatology, Peking University School and Hospital of Stomatology, Beijing 100081, P.R. China. [4]Department of Clinical Laboratory, China-Japan Friendship Hospital, Beijing 100029, P.R. China. [5]Department of Dental Materials & Dental Medical Devices Testing Center, Peking University School and Hospital of Stomatology, Beijing 100081, P.R. China. [6]These authors contributed equally: Jia Song, Wei Zhao. [7]These authors jointly supervised this work: Yuxin Yin, Xuehui Zhang, Xuliang Deng, Dan Lu. ✉e-mail: yinyuxin@bjmu.edu.cn; zhangxuehui@bjmu.edu.cn; kqdengxuliang@bjmu.edu.cn; taotao@bjmu.edu.cn

regulated is critical for cancer prevention and also for therapeutic treatment.

As a pattern recognition receptor (PRR), RIG-I specifically recognizes cytosolic short double-strand viral RNA (dsRNA) via its DEAD (Asp-Glu-Ala-Asp) box domain. Subsequently, RIG-I undergoes conformational changes to activate the type I interferon (IFN) signaling cascade, eventually promoting host antiviral immune response[5]. In addition to the key role of RIG-I in the host defense against viruses, RIG-I is also implicated in cancer development and intestinal inflammation[6,7]. Previous study reveals that RIG-I expression is downregulated in human hepatocellular carcinoma (HCC)[8]. Through impairing STAT1 activation, RIG-I deficiency predicts poor outcome of HCC patients and resistance to immunotherapy[8]. It is conceivable that loss of RIG-I is involved in cancer immune escape. However, Rig-i-deficient mice exhibit more severe intestinal inflammation in murine experimental colitis, along with activation of systemic immune response[9]. In spite of a potential association between RIG-I and Gαi2, a subunit of G protein, which is involved in a range of biochemical activities, the underlying mechanisms by which loss of RIG-I triggers persistent nonspecific inflammatory response remain to be defined.

Similar to RIG-I, DEAD-box RNA helicase 3, X-linked (DDX3X) is also one member of DEAD-box protein. DDX3X has been reported to be involved in several signaling pathways, including type I IFN signaling pathway, Wnt/β-catenin signaling pathway and epithelial-mesenchymal transition (EMT) signaling pathway[10]. For instance, DDX3X can act as an adaptor which facilitates RLR signaling pathway through interacting with MAVS, IKKε, and TBK1, consequently leading to type I IFN production and strengthening host antiviral immunity[11,12]. In addition, DDX3X can promote EMT program by suppressing E-cadherin expression and enhance β-catenin trafficking to nuclear and trigger Wnt/β-catenin/TCF signaling, consequently resulting in metastasis of colon cancer[13]. DDX3X thus seems as a potential target for the treatment of cancer. However, the mechanism by which DDX3X activity is regulated remains elusive.

Circular RNAs (circRNAs) are derived from precursor mRNA (pre-mRNA) back-splicing, which is different from the canonical splicing for linear mRNA[14]. There is a competition between circRNAs and their linear mRNA counterparts, which indicates that the presence of circRNAs negatively modulates their parental gene transcription[15]. Physiologically, circRNAs express at low level due to the low efficiency of circRNAs biogenesis, which can be regulated by both cis-acting RNA sequences and RNA-binding proteins[16]. For instance, the DEAH box protein 9 (DHX9) as the nuclear RNA-binding protein that binds specifically to inverted-repeat Alu elements in pre-mRNA, whose deficiency leads to an increase in the amount of circRNAs[17]. In addition to the role in the regulation of transcription, ectopic expression of circRNA can be recognized by PRRs and triggered host immune response[18], but how they act in this context remains to be determined.

In this study, we identify frameshift mutations of RIG-I occurring in patients with colon cancer. Utilizing Rig-i^{fs/fs} mouse model, we find that frameshift mutation of Rig-i increases the susceptibility of mice to colitis and colitis-associated colon cancer. Mechanistic studies show that the frameshift mutation of Rig-i changes the secondary structure of Rig-i pre-mRNA and impairs its association with DHX9, eventually resulting in circRIG-I generation. Moreover, the induction of circRIG-I is required for enhanced oncogenic inflammation in context of Rig-i mutation. Reciprocally, we find that all-trans retinoic acid acts as a DHX9 agonist that limits circRIG-I expression and ameliorates colon inflammatory lesion of Rig-i^{fs/fs} mice. Our data thus uncover the stimulatory role of circRIG-I-DDX3X signaling on non-specific inflammatory response in colon inflammation and its associated cancer development.

## Results

### Germline RIG-I mutations are implicated in colon cancer development

Deregulation of RLR signaling is involved in colonic tumorigenesis[19]. To assess the status of RIG-I/DDX58 in colon cancers, we performed sequencing analysis of the RIG-I exons in 425 blood samples from patients with colon cancer. Comparing RIG-I in patients with 350 healthy control subjects, we identified five patients with heterozygous frameshift variants at residue 395 in exon 3 (three samples with $T^{395} > -$ and two sample with $TTAA^{395-398} > -$), no frameshift variants of RIG-I were detected in healthy control subjects (Supplementary Fig. 1). Since these frameshift mutations introduced a pre-stop codon into the open reading frame (ORF) of RIG-I, we thus hypothesized that this type of germline RIG-I mutation can influence the expression of RIG-I and host innate immunity.

### Generation of mice with frameshift mutant Rig-i

To study the biological function of frameshift variant of RIG-I, we employed CRISPR/Cas9 technology to generate mice with homo- or heterozygous frameshift mutation in exon 3 of Rig-i, which mimics the frameshift variants of RIG-I in human (Fig. 1a). These Rig-i^{fs/fs} mice are fertile and develop normally, which is consistent to the Rig-i^{-/-} mice[9,20]. Subsequent whole exome sequencing and Sanger sequencing further confirmed the fidelity and specificity of Rig-i^{fs/fs} mice (Supplementary Fig. 2 and Supplementary Fig. 3).

To assess the status of RIG-I in Rig-i^{fs/fs} mice, we derived primary mouse embryonic fibroblasts (MEFs) from wild-type (WT), Rig-i^{+/fs}, and Rig-i^{fs/fs} mouse embryos at 13.5-day post-coitum (dpc) and performed western blot with anti-RIG-I antibody against its N-terminal region. As expected, the protein level of RIG-I was undetectable in Rig-i^{fs/fs} MEFs, and weakened in Rig-i^{+/fs} MEFs as relative to those in WT MEFs (Fig.1b). Similar results were also detected in MEFs upon exposure to vesicular stomatitis virus (VSV) (Fig. 1b). In addition, no truncated RIG-I was detected in both Rig-i^{+/fs} and Rig-i^{fs/fs} MEFs (Fig. 1b). To further confirm this result, we employed mass spectrum assay to determine whether the RIG-I gene with frameshift mutation can be translated into protein. Despite many peptides covering the full length of RIG-I were identified in wild-type MEFs, no peptide was detected in Rig-i^{fs/fs} MEFs, which is consistent with the western blot assay (Supplementary Fig. 4).

### Rig-ifs/fs mice exhibit increased susceptibility to colon cancer

To determine whether frameshift mutation of Rig-I is involved in colon cancer development, we used the azoxymethane and dextran sodium sulfate (AOM-DSS) model to induce colitis-associated cancer (CAC). During the inflammatory phase, more pronounced weight loss was detected in Rig-i^{fs/fs} mice as relative to their wild-type littermate controls (Fig. 1c). Moreover, the levels of colon cancer diagnostic markers, including CA19-9 and CEA, were significantly increased in serum from Rig-i^{fs/fs} mice on day 89 post AOM-DSS treatment (Fig. 1d,e). Mice were then euthanized and tumor was evaluated. As shown in Fig. 1f-h, both tumor number and size were dramatically increased in Rig-i^{fs/fs} mice compared with wild-type mice. Histological analysis revealed that more low-grade adenocarcinomas with frequent invasion into submucosa and occasional invasion into muscularis propria were detected in Rig-i^{fs/fs} mice as compared with that in wild-type mice (Fig. 1i). Additionally, we observed that spleens were significantly enlarged in Rig-i^{fs/fs} mice as compared with wild-type mice in the treatment of AOM-DSS (Fig. 1j,k). Unlike the beneficial effects of host anti-tumor immunity, the inflammation plays a detrimental role in colon carcinogenesis[1]. To determine whether the frameshift mutation of Rig-i exerted stimulatory effects on inflammation in colon and facilitates cancer development, we employed the flow cytometry assay to analyze the ratio of immune cells in spleen and lamina propria mononuclear cells (LPMCs) from the tumor area. Compared with lower percentage

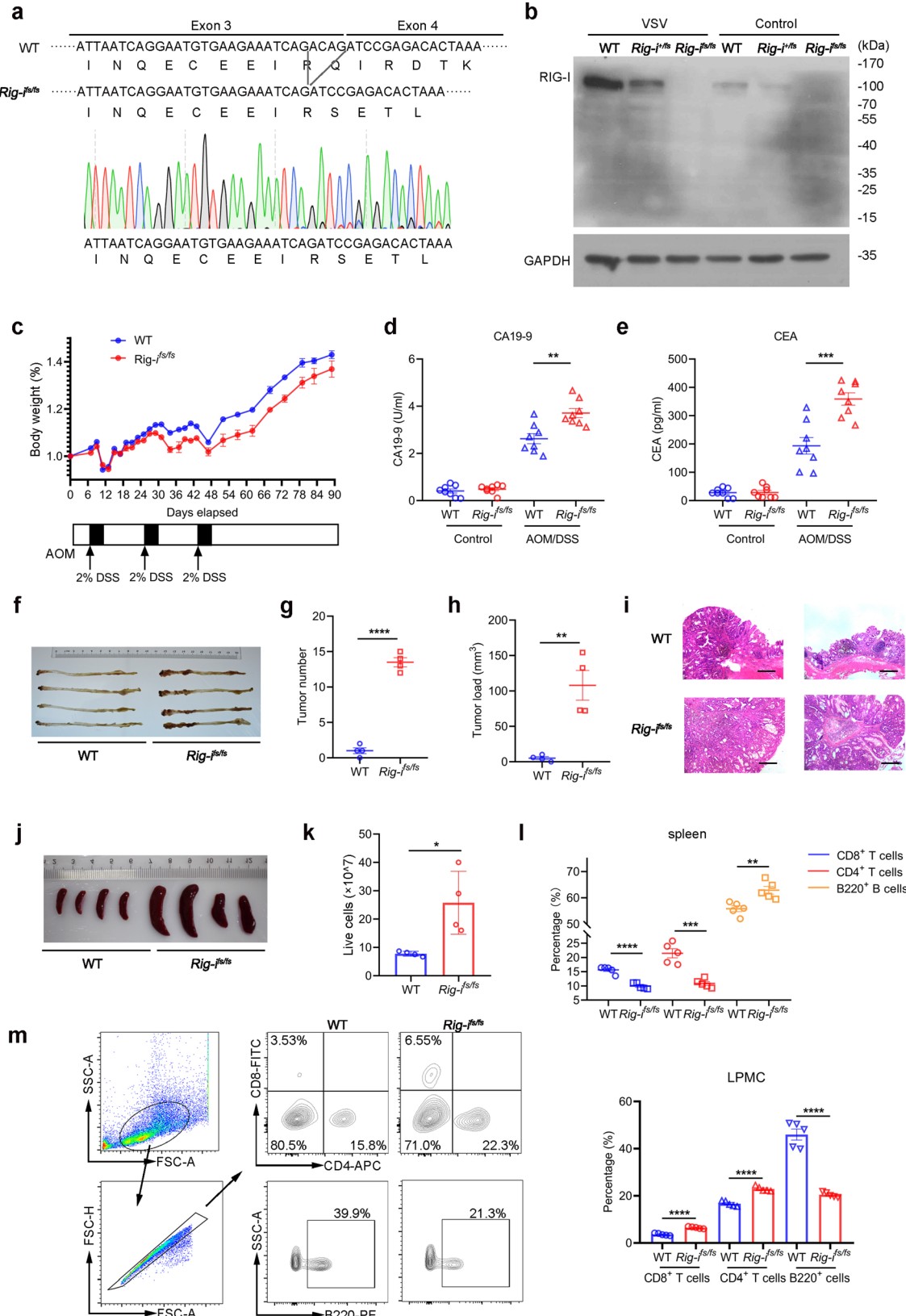

of T cells in spleen, mLN and PPs from *Rig-i*<sup>fs/fs</sup> mice (Fig. 1l and Supplementary Fig. 5), increased amounts of T cells were recruited to the colon from *Rig-i*<sup>fs/fs</sup> mice as compared with wild-type mice in the treatment of AOM-DSS (Fig. 1m). Our data thus indicated that the stimulatory effects of the frameshift mutation of *Rig-i* contribute to colon cancer development.

## Enhanced inflammatory response mainly contributes to the CAC development in Rig-i<sup>fs/fs</sup> mice

To exclude the possibility that the frameshift mutation of *Rig-i* directly promotes tumor initiation, we performed AOM alone and DSS alone experiments in wild-type and *Rig-i*<sup>fs/fs</sup> mice. We monitored the number of tumors induced by AOM without DSS for 6 months and found that

**Fig. 1 | Frameshift mutation of *RIG-I* exacerbates colitis-associated cancer.**
**a** Strategy for generation of *Rig-i^{fs/fs}* mice. The amino acid and nucleic acid sequence of frameshift *Rig-i* mutation were shown. **b** Immunoblot analysis of protein level of RIG-I in MEFs from 6-8-week-old male indicated (wild-type (WT) and *Rig-i^{fs/fs}*) mice with or without VSV infection detected by RIG-I antibody against its N terminal region. Data are collected from 2 independent experiments. **c** The measurement of body weight of 6-8-week-old indicated mice with AOM/DSS treatment (*n* = 8 mice, mean ± s.e.m.). **d-e** The level of CA19-9 (**d**) and CEA (**e**) in indicated mice serum was measured at day 89 in AOM/DSS-induced Colitis-Associated Cancer (CAC) model (*n* = 8 mice, mean ± s.e.m., **P = 0.0020, ***P = 0.0005, two-tailed unpaired Student's t-test). **f** Representative pictures of colon tumors from indicated mice in AOM/DSS-induced Colitis-Associated Cancer (CAC) model. **g-h** The tumor number (**g**) or tumor load (**h**) in the whole colon was measured (*n* = 4 mice, mean ± s.e.m., **P = 0.0028, ****P < 0.0001, two-tailed unpaired Student's t-test). **i** Representative H&E (Hematoxylin-eosin) staining pictures of colon tumors (*n* = 2 mice). The scale bars represent 500 μm. **j-k** Macroscopic evaluation (**j**) and cell number (**k**) of spleen in AOM/DSS-induced Colitis-Associated Cancer (CAC) model (*n* = 4 mice, mean ± s.e.m., *P = 0.0179, two-tailed unpaired Student's t-test). **l-m** Flow cytometric analysis of frequency of CD4⁺ T cell, CD8⁺ T cell and B cell subsets in spleen (*n* = 5 mice, mean ± s.e.m., **P = 0.00544, ***P = 0.0003, ****P < 0.0001, two-tailed unpaired Student's t-test) (**l**) and lamina propria mononuclear cells (LPMC) (*n* = 5 mice, mean ± s.e.m., ****P < 0.0001, two-tailed unpaired Student's t-test) (**m**) isolated from wild-type (WT) and *Rig-i^{fs/fs}* mice treated with AOM/DSS. Source data are provided as a Source Data file.

no differences in the serum levels of CA19-9 and CEA, tumor volume and tumor number between WT and *Rig-i^{fs/fs}* mice, suggesting that the frameshift mutation of *Rig-i* is hardly involved in the tumor initiation (Supplementary Fig. 6).

We next investigated the role of *Rig-i* frameshift mutation in mice upon 2% DSS treatment. As expected, *Rig-i^{fs/fs}* mice exhibited higher mortality, shorter colon length and more severe colonic inflammatory damage than wild-type mice upon DSS treatment (Fig. 2a-d). Moreover, we also measured the protein levels of IL1β and TNFα released into the blood and colon tissue interstitial fluid by ELISA assay. As shown in Supplementary Fig. 7a,b, greater amounts of pro-inflammatory cytokines were released in blood and extracellular matrix in *Rig-i^{fs/fs}* mice as compared with wild-type mice. In addition to acute colitis model, more remarkable pathological lesions were observed after treatment of *Rig-i^{fs/fs}* mice with 2.5% DSS in the acute and recovery phase, as determined by survival curve, weight loss, gross tissue evaluation and colon length (Supplementary Fig. 7c-f). Histological analysis showed that severe crypt distortion, goblet cell depletion as well as massive inflammatory cell infiltration were detected in colons from *Rig-i^{fs/fs}* mice compared with wild-type mice (Supplementary Fig. 7g). Collectively, our data demonstrate that the increased susceptibility of *Rig-i^{fs/fs}* mice to CAC was largely attributed to the stimulatory effects of RIG-I mutation on colon inflammation.

### Rig-i mutation exacerbates inflammatory damage in an epithelial cell-intrinsic manner

To determine whether the enhanced inflammatory response induced by *Rig-i* frameshift mutation is due to the dysregulation of immune cell or nonimmune cell, we employed bone marrow transplantation assay to reconstitute wild-type immune system in *Rig-i^{fs/fs}* mice. Both wild-type and *Rig-i^{fs/fs}* mice were lethally irradiated and transplanted with bone marrow derived from wild-type donor mice. Transplanted mice were left for 2-month to allow for complete reconstitution of the immune system and then subjected to the DSS treatment (Fig. 2e). Following acute DSS colitis induction, *Rig-i^{fs/fs}* mice that received wild-type donor bone marrow exhibited similar increased susceptibility to colitis as the non-transplanted *Rig-i^{fs/fs}* mice did, as determined by survival curve, gross tissue, weight loss evaluation and histological analysis (Fig. 2f-j). Reciprocally, we performed bone marrow transplantation assay to reconstitute wild-type or *Rig-i^{fs/fs}* immune system in wild-type mice. After DSS treatment, the reconstituted mice with *Rig-i^{fs/fs}* donor bone marrow exhibited similar susceptibility to colitis as the mice received with wild-type donor bone marrow did, as determined by weight loss, survival curve and gross tissue evaluation (Supplementary Fig. 7h-j). Our data thus demonstrated that frameshift mutation of *Rig-i* in nonimmune cell rather than immune cell contributes to the severe inflammatory damage.

To decipher the role of RIG-I in host inflammatory response, we performed RNA sequencing (RNA-seq) analysis of colon tissues from transplanted wild-type or *Rig-i^{fs/fs}* mice. Using gene set enrichment analysis (GSEA), we found that frameshift mutation of *Rig-i* facilitated intestinal cell to express higher level of genes associated with type I IFN signaling (Fig. 2k). Given that type I IFN also contributed to the LPS-induced septic shock, we also did an LPS challenge experiment in mice to further confirm the stimulatory role of frameshift *Rig-i* mutation in modulation of inflammatory response. After LPS stimulation, severe mortality was observed in *Rig-i^{fs/fs}* mice as compared with wild-type mice (Fig. 2l). In accordance with the data from DSS-induced colitis model, genes associated with activation of innate immune response were also enriched in livers from *Rig-i^{fs/fs}* mice as compared with wild-type mice (Fig. 2m). Ensued real-time PCR assay was performed to further confirm that higher levels of genes associated with inflammatory response, including *Tnf* and *Isg15*, were detected in livers from *Rig-i^{fs/fs}* mice (Fig. 2n,o). These in vivo results demonstrated that frameshift mutation of *Rig-i* enhances type I IFN signaling in epithelial cell and exacerbates inflammatory tissue damage.

### Viral infection stimulates enhanced immune response by RIG-I mutation

In light of the stimulatory role of *Rig-i* frameshift mutation in regulation of type I IFN, we thus hypothesized that *Rig-i^{fs/fs}* mice were resistant to viral infection. To this end, we employed vesicular stomatitis virus (VSV) to infect wild-type and *Rig-i^{fs/fs}* mice. As shown in Fig. 3a, frameshift mutation of *Rig-i* significantly improved the outcome of mice infected with VSV. Viral replication was inhibited in multiple organs, including lung, spleen and liver from *Rig-i^{fs/fs}* mice as compared with those from wild-type mice (Fig. 3b). Consistent with the data in vivo, we also observed lower viral titers and higher expression of IFN-stimulated genes in *Rig-i^{fs/fs}* MEFs compared with wild-type MEFs MEFs compared with wild-type MEFs (Fig. 3c,d and Supplementary Fig. 8). To further confirm the role of *Rig-i* mutation in the modulation of type I IFN, we used RNA-seq assay to analyze the differentially expressed genes between *Rig-i^{fs/fs}* and wild-type MEFs. As shown in Fig. 3e, genes related with type I IFN signaling were enriched in *Rig-i^{fs/fs}* MEFs as compared with control cells. Our data thus demonstrated that frameshift mutation of *Rig-i* limits viral replication through promotion of type I IFN production.

### Mutation of RIG-I triggers the generation of circular RIG-I

In addition to its protein level, the transcription of *Rig-i* was also downregulated in *Rig-i^{fs/fs}* MEFs, liver as well as spleen with or without virus infection (Fig. 4a-c). Interestingly, we noticed that pre-mRNA of *Rig-i* was increased from *Rig-i^{fs/fs}* mice upon VSV infection in spite of the reduced mRNA level of *Rig-i* (Fig. 4d,e). Given the stimulatory role of RIG-I in the regulation of host innate immunity, it is conceivable that loss of RIG-I hardly stimulates enhanced type I IFN production, which is contradictory to our data that frameshift mutation of *Rig-i* activated IFN signaling. We thus hypothesize that frameshift mutation of *Rig-i* not only affected the expression of RIG-I at post-transcriptional level but also elicited stimulatory effects on the regulation of host immune response.

To this end, we firstly used the RNAfold web server (http://rna.tbi.univie.ac.at/cgi-bin/RNAWebSuite/RNAfold.cgi) to predict RNA secondary structure and found that frameshift mutation significantly

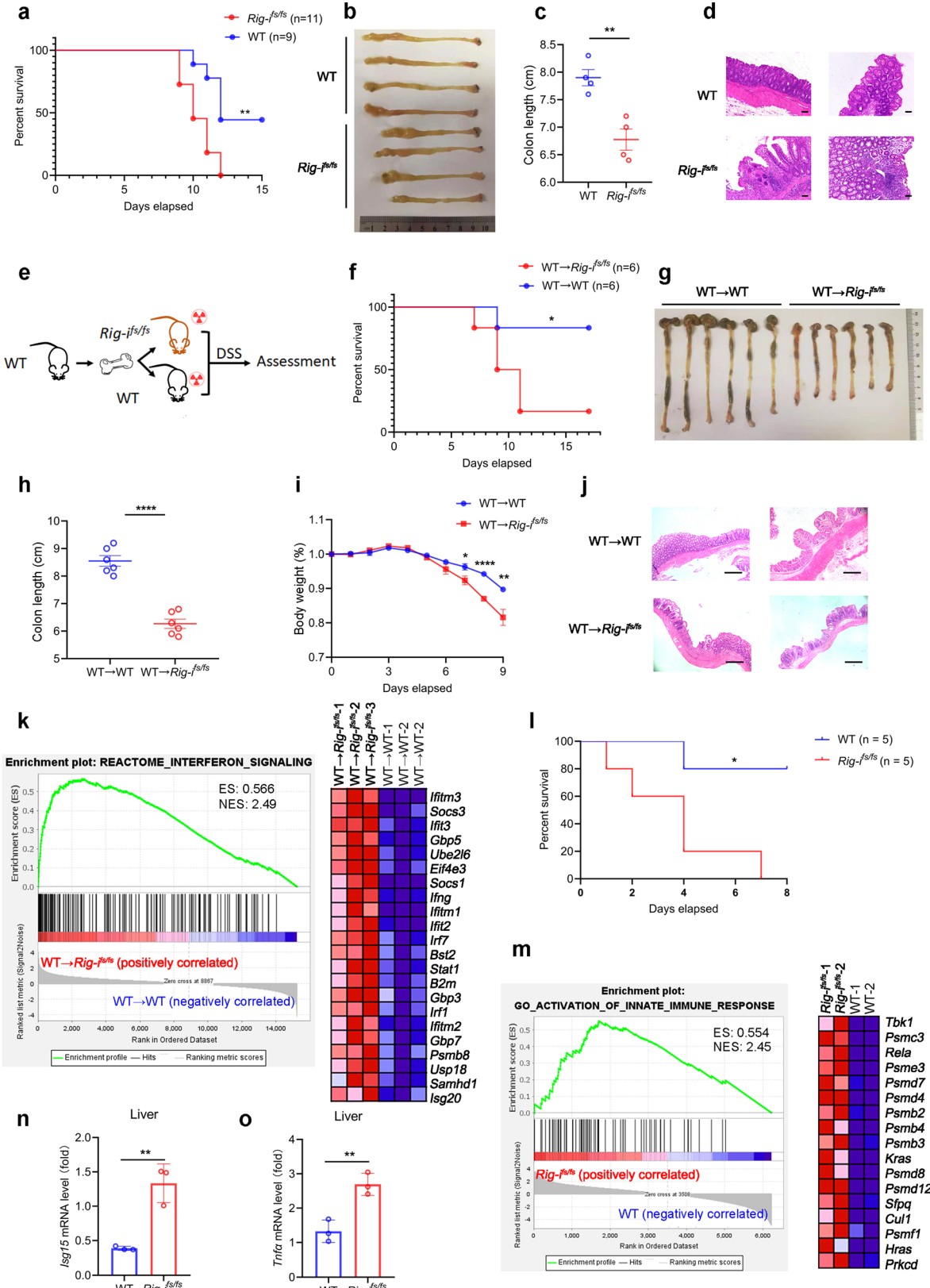

impaired the stem-loop structure relative to that in wildtype RNA (Fig. 4f). Using the published cross-linking immunoprecipitation sequencing (CLIP-seq) data of DHX9[17], we found that the DHX9 peaks on pre-mRNA of *Rig-i* were on or around the short interspersed nuclear elements (SINEs), which is consistent with the preferential enrichment of Alu SINEs among the DHX9 binding sequence (Fig. 4g). Ensued RNA-

pulldown assay further confirmed the relationship between pre-mRNA of *Rig-i* and DHX9 (Fig. 4h). More importantly, the frameshift mutation of *Rig-I* remarkably impaired its association with DHX9 (Fig. 4h). In light of the essential role of DHX9 in modulation of circRNA biogenesis[17], we thus hypothesize that the frameshift mutation of *Rig-I* can trigger circRNA generation from the RIG-I transcript.

**Fig. 2 | *Rig-i^{fs/fs}* mice display increased susceptibility to experimental colitis.**
**a** Survival analysis of described mice (wild-type (WT), *n* = 11 mice; *Rig-i^{fs/fs}*, *n* = 9 mice; **P* = 0.0029, Log-rank (Mantel-Cox) test). **b-c** Macroscopic evaluation (**b**) and colon length (**c**) of indicated mice treated with 2 % (weight/volume) DSS for 8 days (*n* = 4 mice, mean ± s.e.m., **P* = 0.0036, two-tailed unpaired Student's t-test). **d** Representative H&E staining pictures of colon tissues from indicated mice treated with 2 % (weight/volume) DSS (*n* = 2 mice). The scale bars represent 250 μm. **e** A graphic model of bone marrow transplantation assay. **f** Survival analysis of indicated mice transplanted with WT bone marrow in DSS model (*n* = 6 mice, **P* = 0.0326, Log-rank (Mantel-Cox) test). **g-h** Macroscopic evaluation (**g**) and colon length (**h**) of indicated mice transplanted with WT bone marrow in DSS model (*n* = 6 mice, mean ± s.e.m., ****P* < 0.0001, two-tailed unpaired Student's t-test). **i** Body weight (relative to initial weight, set as 100%) of indicated mice transplanted with WT bone marrow in DSS model (*n* = 6 mice, mean ± s.e.m., **P* = 0.0338, ***P* = 0.0073, *P* < 0.0001, two-tailed unpaired Student's t-test). **j** Representative H&E staining pictures of indicated mice colon in DSS model (*n* = 2 mice). The scale bars represent 1000 μm. **k** Gene Set Enrichment Analysis (GSEA) of differentially expressed genes in colon from indicated mice treated with DSS (*n* = 3 mice). ES, enrichment score; NES, normalized enrichment score. **l** Survival analysis of indicated mice (*n* = 5 mice, **P* = 0.0112, Log-rank (Mantel-Cox) test). **m** GSEA of the differentially expressed genes in liver from indicated mice treated with LPS for 24 hours (*n* = 2 mice ES, enrichment score; NES, normalized enrichment score. **n-o** RT-qPCR analysis of the indicated mRNA levels in liver from indicated mice treated with LPS (*n* = 3 mice, mean ± s.e.m., ***P* = 0.0044 (*Isg15*), ***P* = 0.0066 (*Tnfα*), two-tailed unpaired Student's t-test). The primers used for quantitative real-time PCR have been deposited in Supplementary Data 5. Source data are provided as a Source Data file.

In order to reduce the background noise and increase the reliability and accuracy of circRNA identification, we treated ribosomal RNA-depleted total RNAs from virus-infected WT and *Rig-i^{fs/fs}* MEFs with RNase R to degrade linear RNAs and enrich circRNAs, eventually subjected to high-throughput RNA sequencing. Finally, we identified the mmu-DDX58_0004 (circAtlas ID) (http://circatlas.biols.ac.cn/), that was selectively upregulated in *Rig-i^{fs/fs}* MEFs as compared with WT control upon viral infection (Supplementary Fig. 9a). As shown in Fig. 4i and Supplementary Fig. 9b, mmu-DDX58_0004 was derived from exons 5 and 12 of *Rig-i* gene, whose existence can be further confirmed by the specific divergent primers and validated the predicted splice junction of mmu-DDX58_0004 by ensued sequencing analysis. Given that the mmu-DDX58_0004 is derived from the transcript of *Rig-i*, we thus designated it as circRIG-I. In contrast to its linear counterpart, circRIG-I is abundant and resistant to RNase R treatment (Fig. 4j). Moreover, we used probes that hybridize with the splicing junction to distinguish circRIG-I and probes that hybridize with exon 5 to distinguish circRIG-I and its host gene, RIG-I, by northern blotting (Fig. 4k), which further confirmed that existence of circRIG-I. To analyze the subcellular localization of circRIG-I, we used RNA fluorescence in situ hybridization (FISH) and found that circRIG-I is predominantly localized in cytoplasm (Fig. 4l).

To assess the status of circRIG-I in *Rig-i^{fs/fs}* mice, we used qRT-PCR assay and found that the transcription of circRIG-I was increased in *Rig-i^{fs/fs}* MEFs with virus infection as relative to WT MEFs (Fig. 4m), which is contrast to the reduced mRNA level of *Rig-i* (Fig. 4a). Furthermore, we also analyzed the expression of circRIG-I in vivo. As shown in Fig. 4n, compared with WT mice, circRIG-I was upregulated in lung and liver from *Rig-i^{fs/fs}* mice upon exposure to virus. Our data thus demonstrated that the frameshift mutation of *Rig-i* triggers the production of circRIG-I.

## CircRIG-I is upregulated in colon cancer

Conserved with murine circRIG-I, human circRIG-I can be interrogated by the CIRCpedia V2 database. Ensued sequencing assay further validated the predicted splice junction of human circRIG-I (Fig. 5a). We next measured the expression of circRIG-I in colon cancer tissues and their matched adjacent normal tissues by RT-qPCR (Supplementary Data 1). As shown in Fig. 5b, circRIG-I was upregulated in colon cancers. Through analysis of the clinic-pathological features, we found that ulcerative colon cancers expressed higher level of circRIG-I than polypoid colon cancers did (Fig. 5c). Moreover, upregulation of circRIG-I was highly related with tumor (III/IV) stage and patients' age, while no significant correlation was observed concerning patients' gender, tumor subtype and location (Fig. 5d-h). In addition to colon cancers, we also analyzed the expression of circRIG-I in the inflammatory foci (P) and matched adjacent normal tissues (N) from patients with ulcerative colitis. As shown in Fig. 5i, the expression of circRIG-I was increased in the inflammatory foci as compared with normal tissues, which further support the notion that circRIG-I is involved in tumor-related inflammation. These findings thus indicated that dysregulation of circRIG-I may contribute to colon cancer progression.

## CircRIG-I strengthens host innate immune response

To determine whether circRIG-I is essential for the increased production of type I IFN, we used short hairpin RNA (shRNA) that targets the back-splice junction of circRIG-I to selectively knockdown circRIG-I without affecting its linear counterpart *Rig-i*. As shown in Supplementary Fig. 10a, the expression of circRIG-I rather than its counterpart linear mRNA was downregulated in cell transfected with pLKO-circRIG-I-2 or pLKO-circRIG-I-3 rather than pLKO-circRIG-I-1 or pLKO-circRIG-I-4. We thus employed pLKO-circRIG-I-2 and pLKO-circRIG-I-3 to further analyze the role of circRIG-I in modulation of host antiviral immunity. As shown in Fig. 6a, silence of circRIG-I to some extent impaired the stimulatory effects on the production of type I IFN and ISG (*Ifitm2*) in WT or *Rig-i^{fs/fs}* MEFs upon exposure to VSV. We next transfected circRIG-I into HEK293T cells. PCR result confirmed the enforced expression of circRIG-I in HEK293T cells (Fig. 6b). As shown in Fig. 6c-e, overexpression of circRIG-I restricted viral replication and enhanced type I IFN production. Likewise, the expression of other ISGs was also increased in circRIG-I-expressing cells during viral infection (Fig. 6f).

As the transcriptional factor, the activity of IRF3 is essential for host innate immunity. Canonically, IRF3 is activated through phosphorylation by TBK1, which in turn forms homodimers, eventually resulting in nuclear translocation and initiating the transcription of type I IFNs and various antiviral genes via binding to the target genome region. As shown in Fig. 6g, viral infection triggered IRF3 phosphorylation in both circRIG-I-expressing and control cells. Notably, the phosphorylation level of IRF3 was significantly increased in circRIG-I-expressing cells (Fig. 6g). We next determined whether circRIG-I influenced the dimerization and ensued nuclear translocation of IRF3. Following VSV infection, increased dimerization and nuclear accumulation of IRF3 were detected in circRIG-I-expressing cells as compared with that in control cells (Fig. 6h and Supplementary Fig. 10b). Taken together, our data identified the stimulatory effects of circRIG-I in the modulation of host innate immunity.

## CircRIG-I triggers the activation of DDX3X/MAVS/TRAF5/TBK1 pathway

In response to viral dsRNA, RLRs like RIG-I and MDA5 trigger innate immune response by sequentially activating MAVS/TBK1/IRF3 axis for type I IFN production. Reciprocally, IFN-IFNR signal activates the JAK-STAT pathway, leading to the expression of ISGs, which modulate the program of host immunity. To determine the role of circRIG-I in modulation of host innate immunity, we first transfected circRIG-I into HEK293T cells in presence or absence of IFNβ. As shown in Supplementary Fig. 11a, enforced expression of circRIG-I hardly affected the expression of ISGs under IFNβ treatment. We next co-transfected active truncation of MDA5 (MDA5_{1-350}) or MAVS with circRIG-I into HEK293T cells. Compared with the control cells, overexpression of

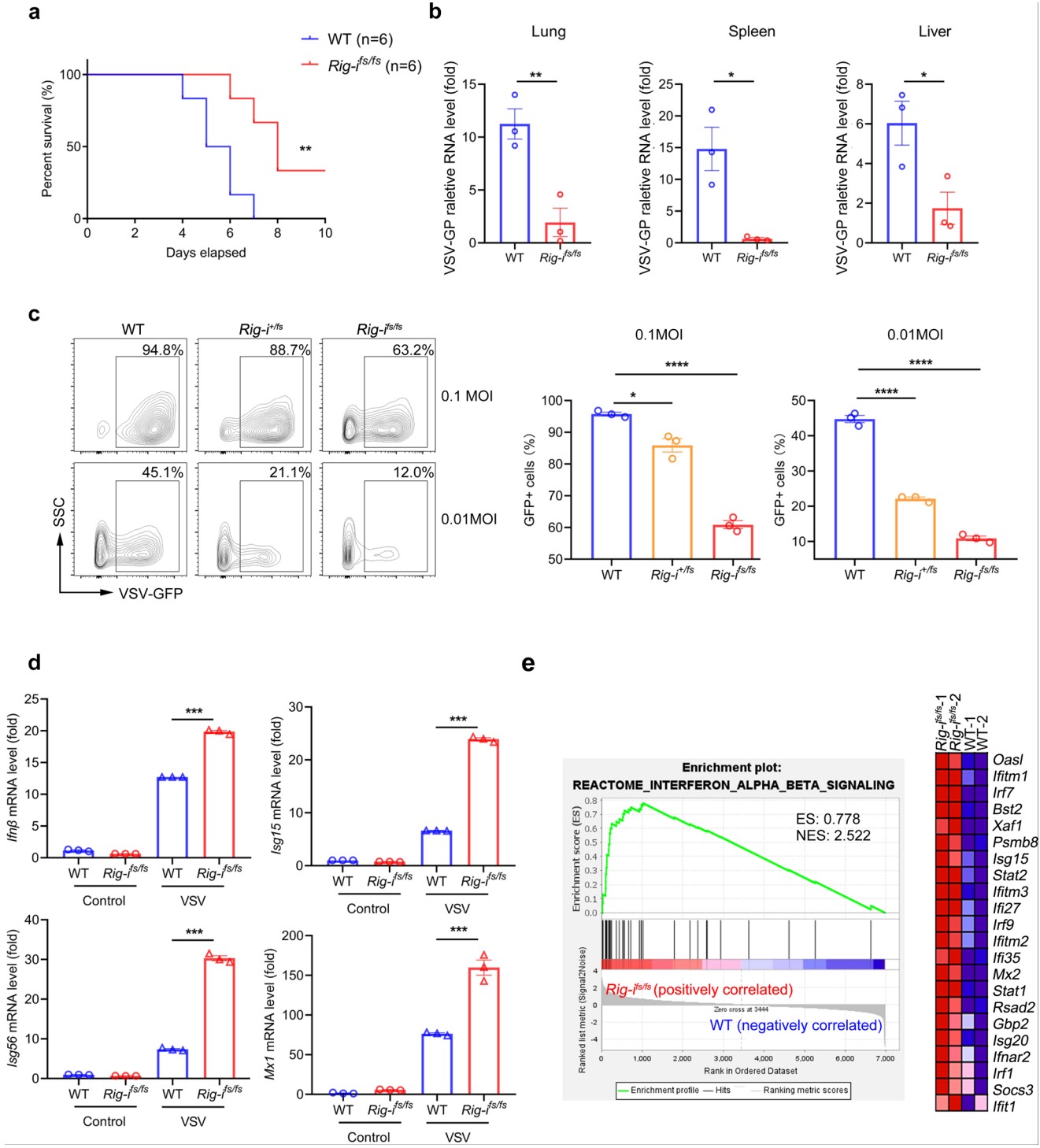

**Fig. 3 | Enhanced immune response by *RIG-I* mutation limits viral replication.**
**a** Survival analysis of wild-type (WT) and *Rig-i^{fs/fs}* mice treated with 1 × 10^8 p.f.u. of VSV (*n* = 6 mice, **P = 0.0062, log-rank (Mantel−Cox) test). **b** RT-qPCR analysis of viral mRNA level in lung, spleen and liver from wild-type (WT) and *Rig-i^{fs/fs}* mice on day 2 post-VSV infection (*n* = 3 mice, mean ± s.e.m., *P = 0.0145 (spleen), *P = 0.0356 (liver), **P = 0.0091, two-tailed unpaired Student's t-test). **c** Flow cytometric analysis of GFP^+ wild-type (WT), *Rig-i^{+/fs}* and *Rig-i^{fs/fs}* MEFs in the treatment of VSV-GFP (*n* = 3, mean ± s.e.m., *P = 0.0112, ****P < 0.0001), two-tailed unpaired Student's t-

test). **d** RT-qPCR analysis of mRNA levels of *Ifnβ* and *ISGs* in wild-type (WT) and *Rig-i^{fs/fs}* MEFs with treatment of VSV (*n* = 3, mean ± s.e.m., ***P < 0.0001, two-tailed unpaired Student's t-test). **e** GSEA of the differentially expressed genes in wild-type (WT) and *Rig-i^{fs/fs}* MEFs treated with VSV for 24 hours (*n* = 2, ES, enrichment score; NES, normalized enrichment score. The primers used for quantitative real-time PCR have been deposited in Supplementary Data 5. Source data are provided as a Source Data file.

circRIG-I strengthened the upregulation of IFNβ and other ISGs induced by RLR signal (Supplementary Fig. 11b). Our data thus indicated that circRIG-I triggers the activation of RLR signaling rather than IFN signaling to suppress viral replication.

To investigate the molecular mechanism by which circRIG-I positively modulates host innate immunity, we developed an MS2-MCP-TurboID system and performed proximity-based labeling screening. Four copies of MS2 was ligated into the vector expressing

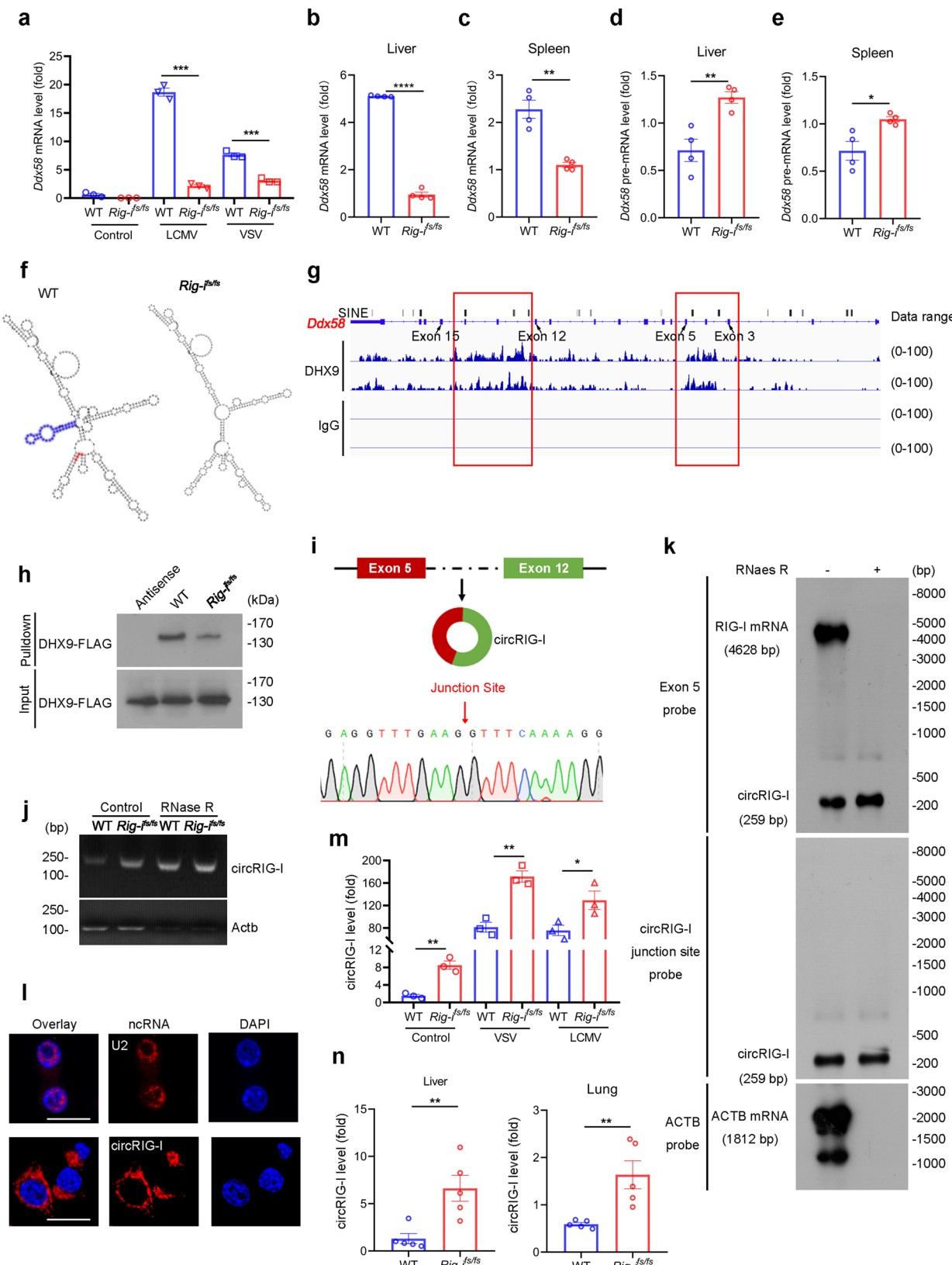

circRIG-I. To determine whether MS2 ligation influences the biological function of circRIG-I, we transfected the circRIG-I or circRIG-I-MS2 into HEK293T cells. Upon exposure to VSV-GFP infection, circRIG-I-MS2 elicited similar antiviral effects of circRIG-I, which further confirmed the availability of circRIG-I-MS2 (Supplementary Fig. 12a). We next co-transfected circRIG-I-MS2 with the vector encoding MS2 coating

protein (MCP)-TurboID into HEK293T cells and treated cells with VSV. Following biotin-affinity capture, the biotinylated proteins were purified and analyzed by mass spectrometry (Fig. 7a). As shown in Supplementary Fig. 12b-d, proteins containing the nucleotide-binding domain or RNA recognition motif were enriched in the interactome of circRIG-I. Among them, a notable finding was the identification of

**Fig. 4 | Frameshift mutation of RIG-I triggers circRIG-I generation. a** RT-qPCR analysis of *Ddx58* mRNA in virus-infected WT and *Rig-i*^fs/fs^ MEFs (*n* = 3 cell cultures, mean ± s.e.m., *** *P* < 0.0001, two-tailed unpaired Student's t-test). **b-e** RT-qPCR analysis of *Ddx58* mRNA (**b-c**) and pre-mRNA (**d-e**) in indicated mice infected with VSV (*n* = 4 mice, mean ± s.e.m., ****P* < 0.0001, ***P* = 0.001046 (**c**), **P* = 0.0182, ***P* = 0.0058 (**d**), two-tailed unpaired Student's t-test). **f** RNA secondary structures of *RIG-I* pre-mRNA (exon 3 and intron 3) were predicted by RNAfold web server. Frameshift mutation and pre-mRNA structural change were highlighted in red and blue, respectively. **g** Integrative Genomics Viewer analysis of CLIP-seq coverage, DHX9 peaks at Alu elements of *Ddx58* were highlighted. **h** Interaction between DHX9 and WT or frameshift mutant *RIG-I* pre-mRNA. Data are collected from 2 independent experiments. **i** Schematic diagram illustrated the formation of circRIG-I (mmu-DDX58_0004) originated from *Rig-i* pre-mRNA (exon 5 and 12). Sequencing analysis of circRIG-I junction site was shown. Data are collected from 2 independent experiments. **j** PCR analysis of circRIG-I in WT and *Rig-i*^fs/fs^ MEFs treated by RNase R or not with specific divergent primers. Data are collected from 2 independent experiments. **k** Northern blotting analysis of circRIG-I and RIG-I mRNA levels in MEFs by hybridization with exon 5 (upper) and exon 5-exon 12 junction (lower) probes with and without RNase R treatment. Data are collected from 2 independent experiments. **l** Subcellular localization of circRIG-I in iBMDM by fluorescence in situ hybridization. The scale bars represent 20 μm. Data are collected from 2 independent experiments. **m** RT-qPCR analysis of circRIG-I expression in indicated MEFs with VSV or LCMV Cl13 infection (*n* = 3 cell cultures, mean ± s.e.m., **P* = 0.045, ***P* = 0.002226 (Control), ***P* = 0.002421 (VSV), two-tailed unpaired Student's t-test). **n** RT-qPCR analysis of circRIG-I expression in indicated tissues from indicated mice infected with VSV (*n* = 5 mice, mean ± s.e.m., ***P* = 0.006939 (Liver), ***P* = 0.008278 (Lung), two-tailed unpaired Student's t-test). Source data are provided as a Source Data file.

DDX3X as a strong circRIG-I binding partner (Fig. 7b and Supplementary Data 2), which was confirm by RNA pulldown assay and confocal assay (Fig. 7c,d). Subsequent cross-linking immunoprecipitation (CLIP) assay demonstrated the linkage of DDX3X with the circRIG-I (Fig. 7e), which further confirmed a physical association between DDX3X and circRIG-I.

To determine whether DDX3X is required for circRIG-I signaling, we thus used shRNA to knockdown of the endogenous MDA5, MVAS, DDX3X or IRF3 in both wild-type and *Rig-i*^fs/fs^ MEFs, respectively (Supplementary Fig. 13a). Following infection with VSV-GFP, although loss of MAVS partly impaired the antiviral effects of *Rig-i*^fs/fs^ MEFs, the lower percentage of GFP⁺ *Rig-i*^fs/fs^ MEFs was observed when endogenous MDA5 was silenced as compared with wild-type MEFs (Supplementary Fig. 13b). Moreover, identical high viral titration and low level of ISGs were detected in both wild-type and *Rig-i*^fs/fs^ MEFs when endogenous DDX3X or IRF3 were knocked down (Supplementary Fig. 13c). Our data thus demonstrate that DDX3X-IRF3 signaling is essential for the antiviral effects of *Rig-i*^fs/fs^ MEFs.

To investigate the mechanisms that circRIG-I promotes DDX3X activation, we used immunoprecipitation followed by mass spectrometry (MS) to identify the dynamic change of DDX3X-associated proteins in presence or absence of circRIG-I (Fig. 7f and Supplementary Data 3). Utilizing the cytoscape, we identified multiple DDX3X-associated proteins, including TBK1, EIF4G1 and EIF3A, which is consistent with previous reports (Fig. 7g, Up). Notably, presence of circRIG-I promoted the association of DDX3X with MAVS, the TRAF family (TRAF1, TRAF4 and TRAF5) as well as TBK1 (Fig. 7g, Down), which are crucial for activation of host innate immunity. To further confirm these results, we co-transfected DDX3X with MAVS, TRAF2, TRAF4, TRAF5, TRAF6 or TBK1 into HEK293T in presence or absence of circRIG-I. As shown in Fig. 7h, the association of DDX3X with MAVS, TRAF4, TRAF5 and TBK1 rather than TRAF2 or TRAF6 was strengthened in presence of circRIG-I. Collectively, our data demonstrated that circRIG-I stimulates IRF3-mediated innate immune signaling through activating DDX3X.

**All-trans retinoic acid ameliorates immunopathology through suppression of circRIG-I expression**

The all-trans retinoic acid (ATRA) has been reported to be involved in regulation of RIG-I expression[21]. Interestingly, we also found that ATRA treatment also promoted *Dhx9* expression that is negative regulation of circRIG-I biogenesis (Fig. 8a). To determine whether ATRA also affects the post-transcription of *Rig-i*, we assessed both the pre-mRNA and mRNA level of *Rig-i* in wild-type MEFs under ATRA treatment. As shown in Fig. 8b, the *Rig-i* mRNA level rather than its pre-mRNA level was gradually upregulated along with the increase of ATRA dose. Reciprocally, the expression of circRIG-I was inhibited in this process (Fig. 8c). Furthermore, ATRA treatment weakened the upregulation of circRIG-I in the context of frameshift mutation of *Rig-i* (Fig. 8d). Given that circRIG-I upregulation is essential for the enhanced immune response in *Rig-i*^fs/fs^ mice, we thus hypothesize that supplementation of ATRA ameliorates the pathological lesion in *Rig-i*^fs/fs^ mice treated with DSS. To this end, we treated wild-type or *Rig-i*^fs/fs^ mice with or without ATRA in presence of DSS. In spite of mild remission of pathological lesions in wild-type mice, supplementation of ATRA remarkably improved the outcome of *Rig-i*^fs/fs^ mice bearing acute DSS-induced colitis, as determined by survival curve, colonic length and histological analysis (Fig. 8e-h).

Collectively, our data demonstrate that frameshift mutation of *Rig-i* triggers circRIG-I, which in turn activates DDX3X/MAVS/TRAF5/TBK1 axis, eventually strengthening type I IFN production and colon inflammatory damage. Reciprocally, ATRA acts as a suppressor of circRIG-I that can be used as a potential strategy to treat severe colon inflammation induced by frameshift mutation of *RIG-I* (Supplementary Fig. 14).

## Discussion

Accumulative studies reveal that RIG-I participates in various cellular activities in addition to detection of exogenous viral RNAs[6]. Here, we identified frameshift *RIG-I* variants in patients with colon cancer. The incidence of *RIG-I* mutation was significantly higher than the population frequency (0.0004%) interrogated from GnomAD database. Moreover, this *RIG-I* mutation has been recorded in the COSMIC database (Mutation ID: COSM6995308), while the function of this mutation is still unknown. Interestingly, this frameshift variant (rs760088776) has also been detected in patients with Singleton-Merten syndrome (SMS). As an autosomal-dominant multi-system disorder, SMS is characterized by dental dysplasia, skeletal abnormalities and other phenotypes[22]. Recent study reveals that SMS is considered as a type I interferonopathy in which an upregulation of type I IFN is considered to have a central role in disease pathogenesis[23]. In other words, frameshift mutant *RIG-I* can stimulate the activation of IFN signaling. Considering the fact that both mRNA and protein level of RIG-I are impaired in context of mutant *Rig-i*, the only plausible explanation is that the inducible circRIG-I exerts positive effects on type I IFN production. Accordingly, our data show that circRIG-I directly interacts with DDX3X and activates the DDX3X signaling, eventually augmenting the inflammatory response through upregulation of type I IFN production. Our data thus provide insight into mutant *RIG-I* action during colitis and other autoimmune disorders.

Previously, two types of *Rig-i* deficient mice were generated, in which exons 8 to 10 (Δ8-10) or exons 4 to 8 (Δ4-8) of *Rig-i* were deleted, respectively, while the phenotype of these mice is controversial[9,24]. In this study, we employed CRISPR/Cas9 technology to generate mice with frameshift mutation in Exon 3 of *Rig-i*, which introduced a pre-stop codon into the open reading frame (ORF) of *Rig-i*. Disruption of RIG-I was demonstrated by whole exome sequencing and western blot. In accordance with the *Rig-i* deficient mice (Δ4-8), our homozygous *Rig-i* knockout mice were viable and fertile, which are different from the high lethality of *Rig-i*^-/-^ mice (Δ8-10) during embryogenesis. Similar

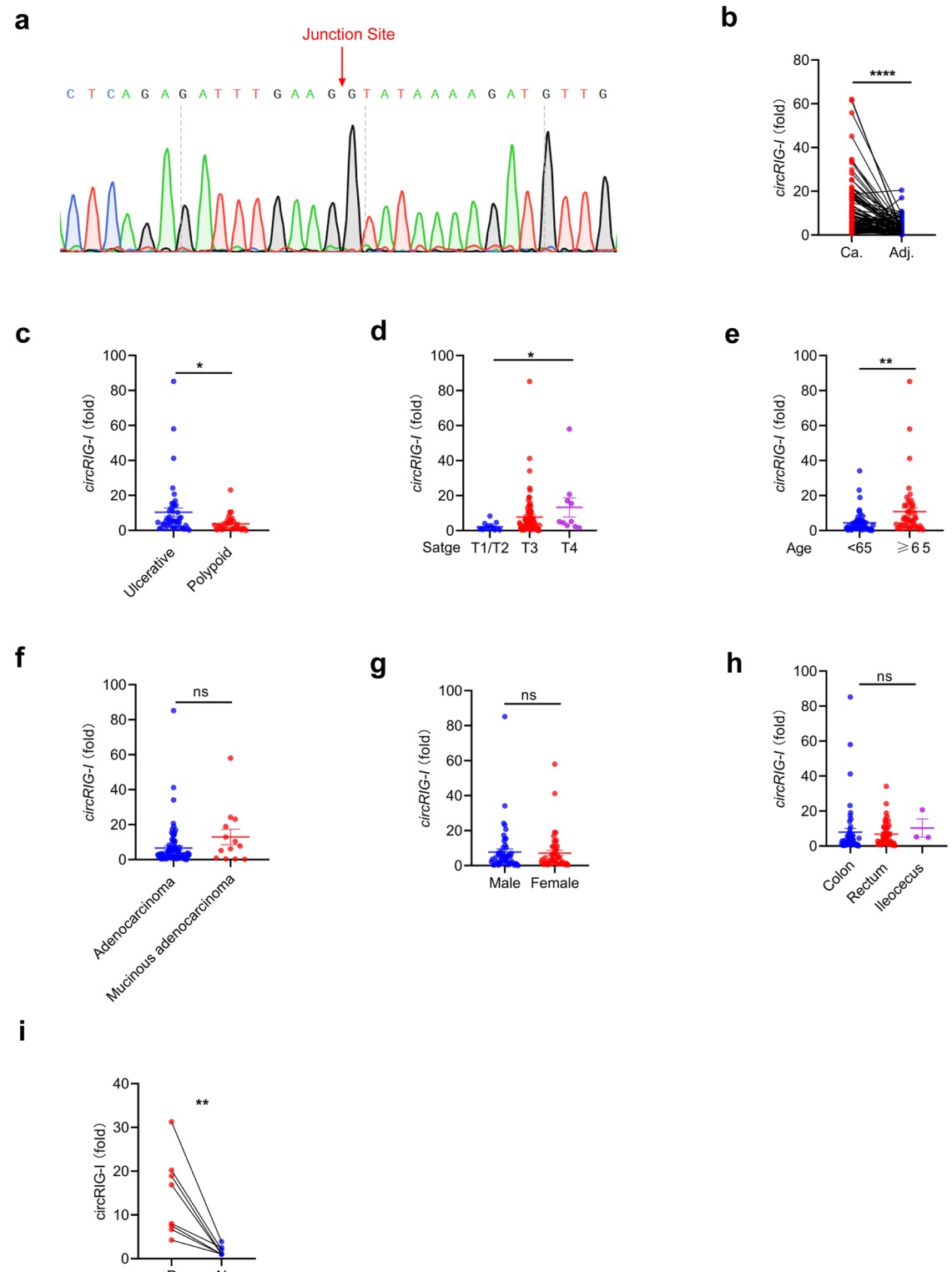

to the increased susceptibility of *Rig-i* deficient mice (Δ4-8) to experimental colitis, *Rig-i*$^{fs/fs}$ mice also exhibited severe colon inflammation and exacerbated malignant transformation of epithelial cells. In addition to colitis, mutant *Rig-i* strengthened inflammatory damage during LPS-induced sepsis and viral infection. In light of the crucial role of inflammation in DNA mutation and immune escape, it is conceivable that the enhanced non-specific inflammation induced by mutant *Rig-i* elicits oncogenic effects in colon.

A pre-mRNA can be spliced to generate a linear or circular RNA. When the pre-mRNA splice sites are joined in the canonical order, a linear mRNA is generated. Otherwise, back-splicing can join a splice donor to an upstream splice acceptor to generate a circular RNA[14]. This competition between canonical splicing and back-splicing helps determine which mature RNAs are generated from a gene[15]. Although the frameshift mutation of *RIG-I* occurred on exon 3 hardly affected the sequences of back-splice/splice sites (Exon 5/Exon 12) involved in back-splicing, it impaired the stem-loop structure relative to that in wildtype RNA. By analyzing published CLIP-seq data, we found that the DHX9 peaks were on exons 3-5 and exons 12-15 of *Rig-i* pre-mRNA, which were on or around the short interspersed nuclear elements (SINEs).

**Fig. 5 | Clinical analysis of circRIG-I expression in colon cancers. a** Sequence of human circRIG-I junction site was shown in the colon cancer sample. Data are collected from 2 independent experiments. **b** RT-qPCR analysis of level of circRIG-I in colon cancers and their matched adjacent normal tissues ($n = 100$ patients, mean ± s.e.m., **** $P < 0.0001$, two-tailed paired Student's t-test). **c** RT-qPCR analysis of level of circRIG-I between ulcerative colon cancer ($n = 44$ patients) and polypoid ($n = 30$ patients) colon cancers (mean ± s.e.m., * $P = 0.0296$, two-tailed unpaired Student's t-test). **d** RT-qPCR analysis of level of circRIG-I in colon cancers with different stage (T1/T2, $n = 14$ patients; T3, $n = 76$ patients; T4, $n = 10$ patients; mean ± s.e.m., * $P = 0.0247$, two-tailed unpaired Student's t-test). **e** RT-qPCR analysis of level of circRIG-I in colon cancers with different age (<65, $n = 52$ patients; ≥ 65, $n = 48$ patients; mean ± s.e.m., ** $P = 0.00635$, two-tailed unpaired Student's t-test). **f** RT-qPCR analysis of level of circRIG-I between colon adenocarcinoma ($n = 87$

patients) and mucinous colon adenocarcinoma ($n = 13$ patients) (mean ± s.e.m., ns, not significant ($P > 0.05$), two-tailed unpaired Student's t-test). **g** RT-qPCR analysis of the relative level of circRIG-I in colon cancers with different gender (male, $n = 53$ patients; female, $n = 47$ patients; mean ± s.e.m., ns, not significant ($P > 0.05$), two-tailed unpaired Student's t-test). **h** RT-qPCR analysis of level of circRIG-I in colon cancers with different locations (colon, $n = 50$ patients; rectum, $n = 47$ patients; ileocecus, $n = 3$ patients; mean ± s.e.m., ns, not significant ($P > 0.05$), two-tailed unpaired Student's t-test). **i** RT-qPCR analysis of level of circRIG-I in the inflammatory foci (P) and matched adjacent normal tissues (N) from patients with ulcerative colitis ($n = 8$ patients; mean ± s.e.m., ** $P = 0.004326$, two-tailed paired Student's t-test). The primers used for quantitative real-time PCR have been deposited in Supplementary Data 5. Source data are provided as a Source Data file.

Considering the key role of the Alu SINEs in cis-regulation of circRNA generation, DHX9 acts as a trans-factor that suppresses RNA processing defects originating from the Alu invasion of the human genome. Our data showed that the frameshift mutation blocked DHX9-binding to the pre-RNA of *Rig-I*, thus releasing the stimulatory effects of Alu elements on back-splicing, eventually resulting in circRIG-I production.

Because of the involvement of DHX9 in multiple cellular processes, including DNA replication, transcription, translation, microRNA biogenesis, RNA processing and maintenance of genomic stability, accumulative studies reveal that DHX9 exerts dual effects on tumor development[25]. Here, we found that ATRA treatment increased the transcription of DHX9, which in turn blocked circRIG-I induction and ameliorated the inflammatory damage in colon. Given that the inflammation plays a detrimental role in colon carcinogenesis[1], our data therefore indicate that, during the inflammatory phase or the early stage of tumorigenesis, increasing DHX9 expression by ATRA treatment ameliorates the intestinal inflammatory damage and blocks tumor progression in a time-dependent manner. In contrast, during the late stage of tumorigenesis, upregulation of DHX9 expression can elicit complex effects on tumor development that may be in a cellular context-dependent manner.

Although many approaches have been developed to analyze the biological function of circular RNA, each of them has its limitation. For instance, chromatin isolation by RNA purification (ChIRP) and RNA pulldown were largely determined by nucleotide binding specificity as well as efficiency[26,27]. In addition, utilizing in vitro transcribed RNA may miss the structure-dependent targets and hardly reveal the transient and dynamic network of circRNA-protein interactions[28,29]. Moreover, MS2-tagged RNA affinity purification (MS2-TRAP) can be affected by excessive or insufficient crosslinking and may not truly reflect physiological conditions[30]. To address the challenging of circRNA-centered interactome mapping, we developed a proximity-based labeling screening combined MS2-MS2 coating protein (MCP) strategy to discover the interaction partners of circRIG-I. By fusing MCP to biotin ligase TurboID, adjacent proteins of MS2 tagged circRNA are covalently labeled so that they can be isolated by streptavidin beads and subsequently identified by mass spectrometry[31]. This circRNA-MS2-MCP-TurboID technology introduced in our study can capture weak and transient circRNA–protein interactions and reveal the signaling circRIG-I involved in. Moreover, ensued CLIP assay further assured the reliability of MS2-MCP-TurboID system. We thus purpose that this effective method could also facilitate future investigation of circRNA-protein interactome.

In summary, our data uncover the stimulatory role of mutant *RIG-I* in colitis and colitis-associated colon cancer development. Through induction of circRIG-I, mutant *RIG-I* activates DDX3X signaling cascade, eventually resulting in type I IFN production and severe inflammatory tissue damage. Reciprocally, supplementation of ATRA ameliorates the cancer-related inflammation by suppression of circRIG-I expression. Thus, our identification of the oncogenic role of circRIG-I provides a potential therapeutic target for treatment of colitis and inflammatory cancer.

## Methods

### Mice
*Rig-I*$^{fs/fs}$ mice (C57BL/6 J background) were generated by CRISPR-Cas9-mediated gene editing. The sequence (GGAATGTGAAGAAATCAGAC) of murine *Ddx58* gene in exon 3 was targeted. All animals were housed and maintained under specific pathogen-free conditions. 6-8-week-old male mice were used. All experiments were performed in accordance with protocols approved by the Ethics Committee of Peking University Health Science Center (ECPKUHSC) (LA2021487).

### Detection and analysis of CRISPR-Cas9 off-target
Genomic DNA was extracted from colon tissue of wild-type and *Rig-I*$^{fs/fs}$ mice. To detect if there are any potential off-targets in *Rig-I*$^{fs/fs}$ mice, we selected the top ten ranked genes that may be misrecognized through Off-Spotter (https://cm.jefferson.edu/Off-Spotter/) and examined their corresponding genome by DNA sequencing (all primers, Supplementary Data 4). Neither genome mutation nor deletion of the corresponding gene was detected in *Rig-I*$^{fs/fs}$ mice. Furthermore, we performed genome-wide detection and analysis of CRISPR-Cas off-targets by whole exome sequencing (WES).

### Patients and specimens
Human blood specimens were obtained from 425 patients with colon cancer and 350 healthy subjects at the China-Japan Friendship Hospital. Genomic DNA was extracted from blood samples using the TIANamp Blood DNA kit (TIANGEN) and was then used as templates for PCR amplification of the *RIG-I* exon, followed by sequencing analysis. Human colon cancers and their matched normal tissues were obtained from patients undergoing colon cancer surgery at the China-Japan Friendship Hospital. All procedures were approved by the Institutional Review Board of the China-Japan Friendship Hospital (No. 2019-50-Q07), and the informed consent was obtained from all subjects (in accordance with the Helsinki Declaration). The primers used to amplify *RIG-I* exon 3 and subsequently sequencing was as follows: forward 5′-GAAGACCTAAAGTGTTGGCTGAC-3′, reverse 5′- GGGTTTCAATGATCTTTCTCAGGT-3′.

### Induction of CAC (Colitis-Associated Cancer)
For induction of CAC[32], 6-week-old male *Rig-I*$^{fs/fs}$ and wild-type (WT) mice were used. 7 days after a single intraperitoneal injection with 10 mg/kg of Azoxymethane (AOM) (Sigma, A5486-25MG), mice were treated with 2% (weight/volume) dextran sulfate sodium (DSS) (MP, 0216011090) for 3 cycles. 1 cycle includes 7 days DSS treatment and followed 14 days resting (drinking regular water). Mice were sacrificed at day 90 before tumor size reaches the maximal tumor burden (10% of mouse weight or 2000 mm3 at the endpoint) permitted by ECPKUHSC

### Induction and evaluation of acute colitis
For induction of acute colitis, 6-week-old male *Rig-I*$^{fs/fs}$ and wild-type (WT) mice were treated with 2% or 2.5% DSS and body weights were recorded every day.

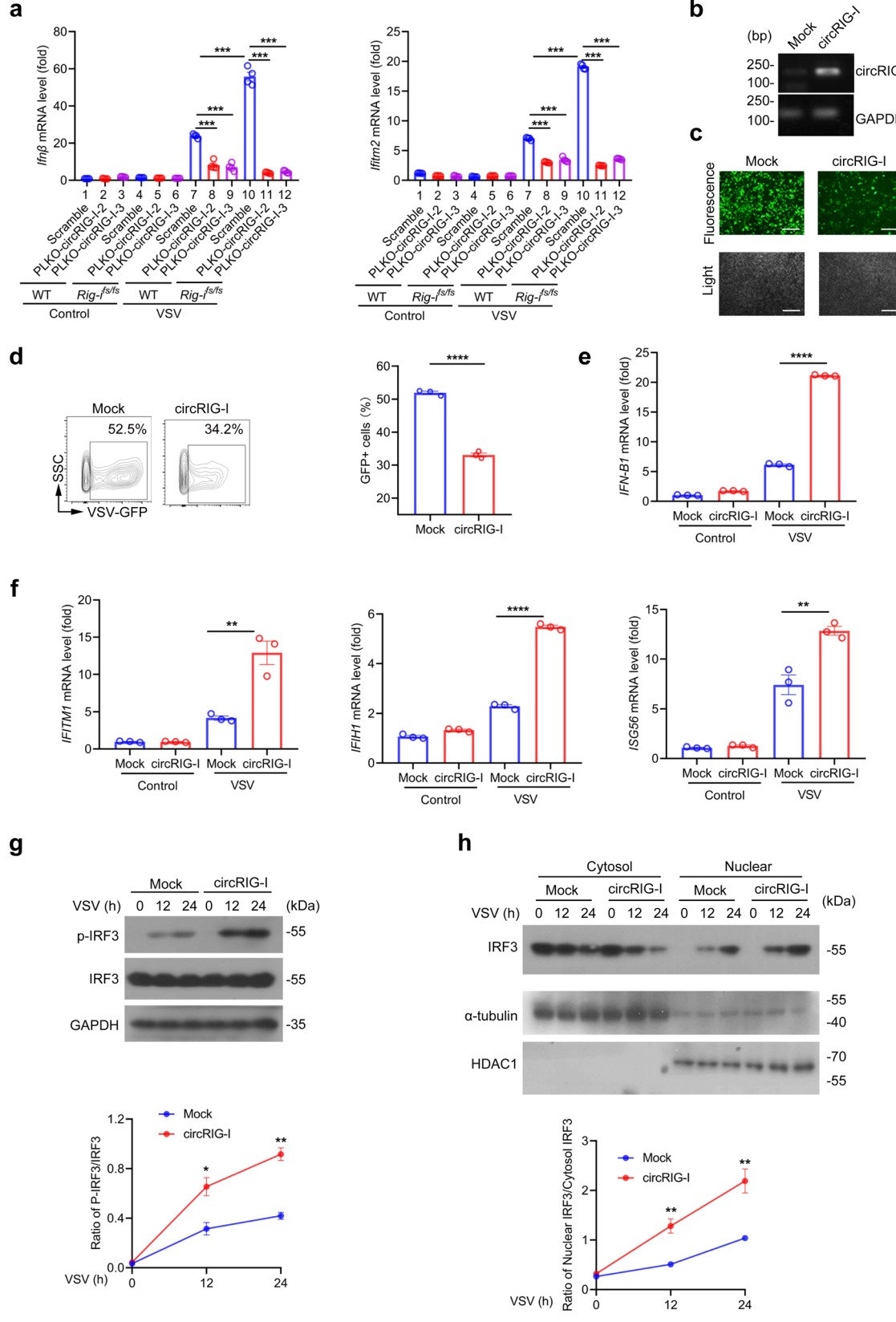

**Nature Communications** | (2022)13:7096

## Bone marrow transplantation

Recipient mice were lethally irradiated (1000 cGy/mouse), and *i.v.* injected with $5 \times 10^6$ bone marrow cells from wild-type or *Rig-i^{fs/fs}* mice femurs and tibias. Experiments on transplanted mice were performed after 2 months to ensure bone marrow reconstitution.

## Enzyme-linked immunosorbent assay (ELISA)

Enzyme-linked immunosorbent assays for CEA and CA19-9 in mouse serum were performed according to the manufacturer's instructions (CEA: Dogesce, DG30128M; CA19-9: MEIMIAN, MM-44828M2).

Tissue interstitial fluid (TIF)[32] was obtained as follows. Intestinal tissue was placed on a 40-µm cell strainer and centrifuged at 40 g for

**Fig. 6 | CircRIG-I enhances host innate immune response. a** RT-qPCR analysis of *Ifnβ* and *Ifitm2* mRNA levels in wild-type (WT) and *Rig-i^{fs/fs}* MEFs in presence or absence of circRIG-I (*n* = 4 cell cultures, mean ± s.e.m., ***P < 0.0001, two-tailed unpaired Student's t-test). **b** Validation of the expression of murine circRIG-I in HEK293T cells by PCR assay. Data are collected from 2 independent experiments. **c** Fluorescence and light field of mock and murine-derived circRIG-I overexpressing HEK293T with VSV-GFP infection. The scale bars represent 200 μm. Data are collected from 2 independent experiments. **d** Flow cytometric analysis of GFP^+ cells from mock and murine circRIG-I overexpressing HEK293T followed infection with VSV-GFP (*n* = 3 cell cultures, mean ± s.e.m., ****P < 0.0001, two-tailed unpaired Student's t-test). **e-f** RT-qPCR analysis of *IFNB* (*n* = 3 cell cultures, mean ± s.e.m., ****P < 0.0001, two-tailed unpaired Student's t-test) (**e**) and ISGs mRNA levels in mock and murine circRIG-I overexpressing HEK293T cells with VSV infection (*n* = 3 cell cultures, mean ± s.e.m., **P = 0.0055 (*IFITM1*), **P = 0.005704 (*ISG56*), ****P < 0.0001, two-tailed unpaired Student's t-test) (**f**). **g** Immunoblot analysis of phosphorylation level of IRF3 in iBMDM overexpressing mock or murine circRIG-I followed with VSV infection (*n* = 3 cell cultures, mean ± s.e.m., *P = 0.0186, **P = 0.0011, two-tailed unpaired Student's t-test). **h** Subcellular localization of IRF3 in iBMDM overexpressing mock or circRIG-I treated by VSV analyzed by nucleus-cytoplasm extraction (*n* = 3 cull cultures, mean ± s.e.m., **P = 0.0065 (12 h), **P = 0.0090 (24 h), two-tailed unpaired Student's t-test). The primers used for quantitative real-time PCR or PCR assay have been deposited in Supplementary Data 5. Source data are provided as a Source Data file.

---

5 min at 4 °C to remove surface liquid. Samples were then spun at 400 g for another 10 min at 4 °C and fluid was collected as TIF. TIF and plasma were spun at 10,000 g for 5 min to remove insoluble particles and the supernatant was assayed for ensued ELISA detection.

ELISA for IL1β and TNFα in mouse serum and tissue interstitial fluid were performed according to the manufacturer's instructions (IL-1β: R&D, MLB00C; TNFα: R&D, MTA00B).

### LPS-induced septic shock
6-week-old male *Rig-i^{fs/fs}* and wild-type (WT) mice were injected intraperitoneally with LPS (10 mg/kg), and their health status was monitored every day.

### VSV infection model
For VSV infection, 6-week old male wild-type or *Rig-i^{fs/fs}* mice were infected by intravenous injection with $1 \times 10^8$ p.f.u. of VSV.

### All-trans retinoic acid (ATRA) treatment
*Rig-i^{fs/fs}* and wild-type (WT) mice were injected treated with 2 % DSS and were injected intraperitoneally with ATRA in corn oil (15 mg/kg) at day 3.

### Co-immunoprecipitation and immunoblot analysis
HEK293T cells were transfected with appropriate plasmids and lysed by co-immunoprecipitation lysis buffer (10% glycerol, 0.5% NP-40, 150 mM NaCl, 0.1 mM EDTA) with protease inhibitor cocktail (Roche). Cell lysates were incubated with anti-FLAG antibody (Sigma, F3165) and protein A/G (Santa Cruz Biotechnology, sc-2003). The beads were washed by PBSN (PBS containing 0.1% NP-40) three times and subjected to SDS-Page. For subcellular fractionation, nuclear and cytoplasmic extracts were isolated with a nuclear-cytoplasmic extraction kit (Applygen, P1200) following the manufacturer's protocol.

Antibodies used in this study were as follows: anti-RIG-I (Santa Cruz Biotechnology, sc-376845, 1:1000), anti-p-IRF3 (Cell Signaling Technology, #4947, 1:1000), anti-MDA5 (Abclonal, A2419, 1:500), anti-MAVS (abcam, ab189109,1:2000), anti-IRF3 (abcam, ab68481, 1:2000), anti-DDX3X (Santa Cruz Biotechnology, sc-365768, 1:500), anti-DHX9 (Santa Cruz Biotechnology, sc-137232, 1:500), anti-GAPDH (RayAntibody, RM2002, 1:5000), anti-α-tubulin (RayAntibody, RM2007, 1:5000), anti-HDAC1(Santa Cruz Biotechnology, sc-8410, 1:1000), anti-FLAG (Sigma, F3165, 1:5000), anti-GFP (RayAntibody, RM1008, 1:5000), anti-GFP (RayAntibody, RM1008), KPL peroxidase-labeled antibody to mouse IgG (H + L) (Seracare, 5220-0341, 1:5000) and KPL peroxidase-labeled antibody to rabbit IgG (H + L) (Seracare, 5220-0336, 1:5000).

### Northern blot
Ten micrograms of total RNA with or without RNase R digestion was resolved on 1.2% agarose gels prepared with formaldehyde and transferred to a Hybond-N + membrane. RNA was fixed to the membrane through UV crosslinking (1200 J). Hybridization was performed at 68 °C overnight with a biotin-labeled probe. The membranes were blocked in blocking buffer for 30 min, and then incubated with streptavidin-HRP for 1 hour. After washed with washing buffer for 3 times, incubated with detection solution for 5 min, the membrane was exposed to X-ray film. The probe sequences were displayed as follows: circRIG-I junction site probe, 5'-Biotin-AGCCCTTTTGAAACCTTCA AACCTCCG-3'; Exon 5 probe, 5'-Biotin- CCTGAATGAAAATCTGGATG CTGGA-3'; Actb probe, 5'-Biotin- CTCTTTGATGTCACGCACGAT-3'

### H&E (Hematoxylin-eosin) staining
For H&E staining, the tissues were fixed with formaldehyde and then embedded with paraffin. Sections of 4 μm thick were used for H&E staining with a standard protocol. Images were acquired using an Olympus IX51 microscope.

### Quantitative real-time PCR
Total RNA was extracted with Trizol reagent (Invitrogen) and then reverse transcribed into cDNA according to the manufacturer's protocol (Promega). The cDNA quantification was done using ABI 7500 Detection System. The sequences of the PCR primers are listed in Supplementary Data 5.

### Flow cytometry
To detect viral replication, HEK293T, MEFs and iBMDM were infected with VSV-GFP, and GFP^+ cells were measured by flow cytometry.

To analysis immune cell population in lymph nodes, spleen and LPMC[32], lymphocytes were isolated and incubated with specific antibodies for 30 min at room temperature. APC-labeled anti-CD4 antibody (Biolegend, GK1.5, 1:500), FITC-labeled anti-CD8 antibody (Biolegend, 53-6.7, 1:500) and PE-labeled anti-B220 antibody (Biolegend, RA3-6B2, 1:500) were used.

### RNA sequencing analysis
For study of the role of RIG-I deficiency in antiviral immunity, mouse embryonic fibroblasts (MEFs) were prepared from 13.5-day-old embryos derived from wild-type or *Rig-i^{fs/fs}* mouse and cultured in vitro treated by VSV or LCMV Cl13 for 24 hours. Total RNA was then purified using poly-T oligo-attached magnetic beads. RNA-seq libraries were constructed using NEBNext® UltraTM RNA 24 Library Prep Kit for Illumina® (NEB, USA) and sequenced on an Illumina platform with 125 bp/150 bp paired-end reads. According to the manufacturer's instructions, clean data were obtained by removing reads containing adapter and poly-N as well as low quality reads from raw data. Clean reads were mapped with the reference genome Hisat2 (version 2.0.5) based on the gene model annotation file. Fragments per kilobase per million mapped reads (FPKM) of each gene was calculated based on the length of the gene and reads count mapped to this gene. Gene sets from RNAseq data were analyzed for overlap with curated datasets (C5.all.V6.2, H.all.V6.2) in the MSigDB using the web interface available at http://software.broadinstitute.org/gsea/index.jsp.

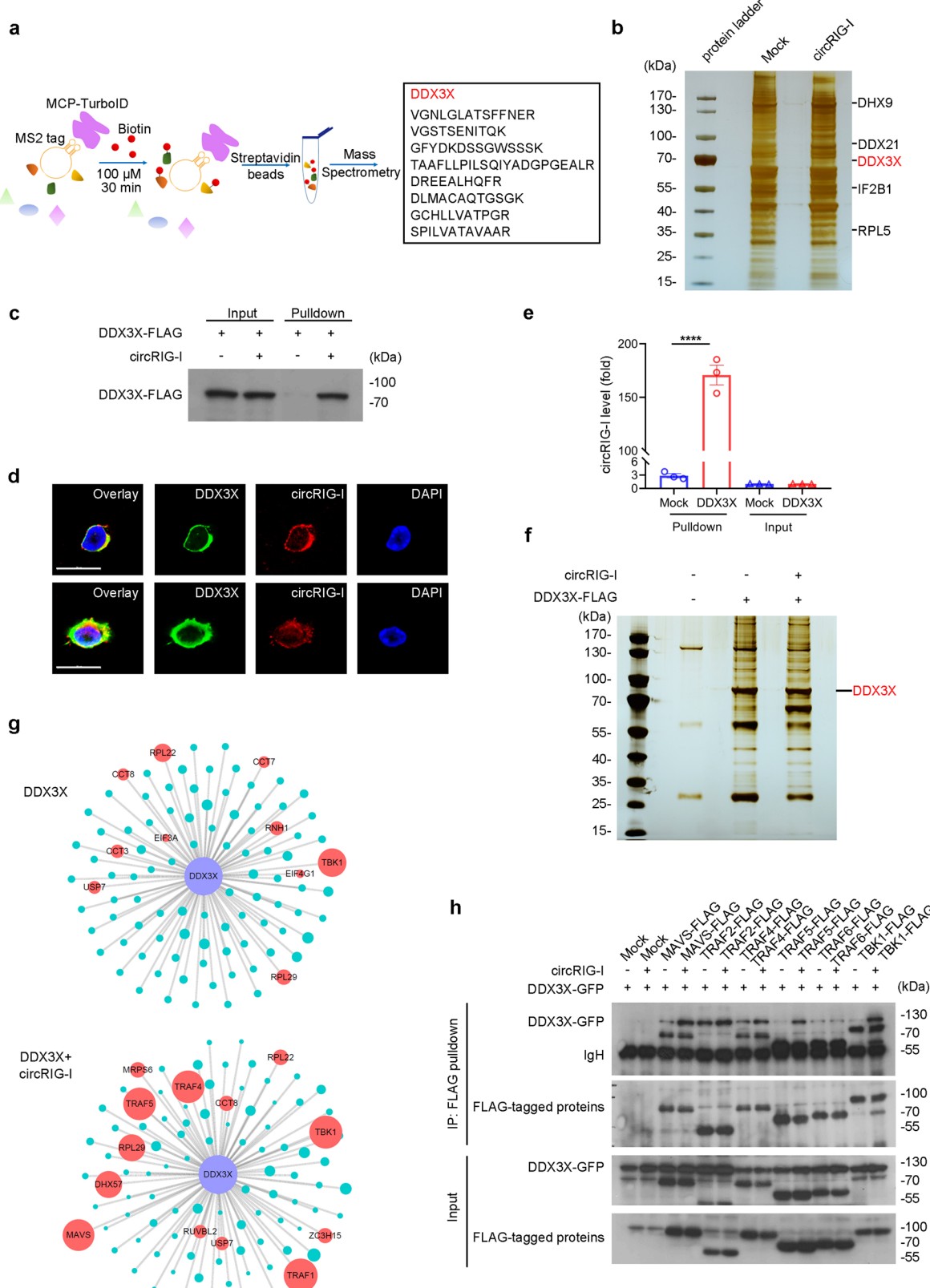

## Cells

HEK293T cells were from American Type Culture Collection (ATCC). Primary mouse embryonic fibroblasts (MEFs) from wild-type (WT), *Rig-i*[+/fs] or *Rig-i*[fs/fs] mouse embryos at 13.5-day post-coitum (dpc). Immortalized bone-marrow-derived macrophages (iBMDMs) were provided by Dr. Fuping You (Peking University Health Science Center, China).

To obtain iBMDM stably transfected cells, sequence of circRIG-I was cloned into pLV-ciR lentiviral vector. HEK293T cells were transfected with psPAX2, pMD2.G and lentiviral constructs. 48 hours after transfection, supernatants supplemented with 8 μg/ml polybrene were used for infection. In order to get overexpressing stably cells, 2 μg/ml puromycin was used to search out positive cells.

**Fig. 7 | CircRIG-I activates DDX3X-mediated innate immune signaling.**
**a** Workflow diagram of circRNA-MS2-MCP-TurboID system. MS2 is a stem-loop bacteriophage RNA that specifically binds to the MS2 coating protein (MCP) with high affinity. By fusing MCP and biotin ligase TurboID, adjacent proteins of MS2 tagged circRIG-I are covalently labeled by biotin. Followed by streptavidin beads enrichment of biotinylated proteins and mass spectrometry, circRIG-I-associated proteins can be identified. **b** Mass spectrum analysis of circRIG-I-associated proteins according to the workflow shown in (**a**). Data are collected from 2 independent experiments. **c** Vectors encoding DDX3X-FLAG and murine-derived circRIG-I were co-transfected into HEK293T cells. After 24 hours of transfection, RNA pulldown assay was performed by biotin labeled nucleotide probe specific to circRIG-I. Data are collected from 2 independent experiments. **d** Confocal examination of the colocalization between DDX3X and circRIG-I in iBMDM cells. The scale bars represent 20 μm. Data are collected from 2 independent experiments.

**e** HEK293T cells were transfected with vectors encoding DDX3X-FLAG or murine circRIG-I. UV-CLIP assay was performed and followed RT-qPCR analysis verified the protein-circRNA association (*n* = 3 cell cultures, mean ± s.e.m., ns, ****$P$ < 0.0001, two-tailed unpaired Student's t-test). **f** Mass spectrum analysis of DDX3X-associated proteins in presence or absence of circRIG-I. The band indicating DDX3X was shown. Data are collected from 2 independent experiments. **g** Protein-protein interaction (PPI) analysis of DDX3X interactome in presence or absence of circRIG-I by Cytoscape Microsoft. **h** HEK293T cells were co-transfected with vectors encoding FLAG tagged proteins and DDX3X-GFP with or without murine-derived circRIG-I. At 24 hours later, cell lysates were immunoprecipitated with FLAG antibody and protein A/G agarose beads and analyzed by immunoblot with anti-GFP antibody. Data are collected from 2 independent experiments.The primers used for quantitative real-time PCR have been deposited in Supplementary Data 5. Source data are provided as a Source Data file.

For circRIG-I knockdown MEFs cells, shRNAs targeting junction site of circ_RIG-I (1#: CTCGGAGGTTTGAAGGTTTCA; 2#: GAGGTTTGA AGGTTTCAAAAG; 3#: GTTTGAAGGTTTCAAAAGGGC; 4#: GAAGGTTT CAAAAGGGCTGAA) were cloned into pLKO.1 plasmid. MEFs were transfected with psPAX2, pMD2.G and lentiviral constructs. Positive cells were selected as mentioned above.

For knockdown MDA5, MAVS, DDX3X and IRF3 in MEFs cells, shRNAs targeting *Ifih1* (1#: CCCATGAGGTATTGTCCTAAA; 2#: CCACA-GAATCAGACACAAGTT), *Mavs* (1#: GCTGAGTCAGAGAAACTTAAA; 2#: GCAACCAGACTGGACCAAATA), *Ddx3x* (1#: CCCTGCCAAACAAGCT AATAT; 2#: GTGGAGTTCTAGTAAAGATAA) and *Irf3* (1#: GCGTCTAGG CTGGTGGTTATT; 2#: TGCGGTTAGCTGCTGACAATA) were cloned into pLKO.1 plasmid. MEFs were transfected with psPAX2, pMD2.G and lentiviral constructs. Positive cells were selected as mentioned above.

### Fluorescence in situ hybridization (FISH) and Immunofluorescence
In situ hybridization was performed with a Fluorescent In Situ Hybridization (FISH) Kit (RiboBio, Guangzhou, China). Briefly, iBMDM cells were fixed with 4% paraformaldehyde, permeabilized with 0.5% Triton X-100 and prehybridized at 37 °C for 30 min. Biotin-labeled FISH probe was added into hybridization buffer at 37 °C overnight. After washed with saline sodium citrate (SSC), cells were incubated with antibody against DDX3X at room temperature or not. Cells were incubated with Alexa Fluor 555-labeled Streptavidin (Bioss, bs-0437P-AF555) or fluorophore-conjugated secondary antibodies and DAPI. Images was acquired with a Nikon TCS A1 microscope. The probe sequences were displayed as follows: circRIG-I probe, 5′-Biotin-AGCCCTTTTGAAACCT TCAAACCTCCG-3′, U2 probe, 5′-Biotin-CTACACTTGATCTTAGCCAA AAGGCCGAGAAGC-3′.

### FLAG pulldown
In brief, HEK293T cells were transfected with indicated plasmids. After 24 hours, cells were lysed with co-immunoprecipitation lysis buffer. FLAG-DDX3X or FLAG-NLS-RIG-I protein was enriched with anti-FLAG M2 beads (Sigma, F2426) at 4 °C for 4 hours. The binding components were eluted with 3×FLAG peptide (Sigma, F4799). The samples were subjected to NuPAGE 4%-12% gel (Invitrogen) and sliver staining (Pierce, 24612). The excised gel segments were subjected to mass spectrum analysis[33,34].

### Proximity biotinylation
HEK293T cells were transfected with plasmids expressing MS2 coating protein (MCP)-TurboID fusion protein and 4 × MS2 tagged circRIG-I. After 24 hours of transfection, biotin was added at a final concentration of 100 μM for 30 minutes. Cells were washed four times with ice-cold PBS before harvesting. Cells were lysed by RIPA (50 mM Tris, 150 mM NaCl, 0.1% SDS, 0.5% sodium deoxycholate, 1% Triton X-100) with protease inhibitor cocktail (Roche) at 4 °C for

30 minutes. Streptavidin agarose beads (Millipore, 69203) were used to enrich biotin-labeled proteins at 4 °C for 3 hours. The beads were subsequently washed with RIPA lysis buffer twice, 1 M KCl once, 0.1 M Na$_2$CO$_3$ once, 2 M urea in 10 mM Tris-HCl (pH = 8.0) once and RIPA lysis buffer twice. Finally, biotinylated proteins were eluted by boiling the beads with elution buffer (55 mM Tris-HCl (pH = 8.0), 0.1% SDS, 6.66 mM DTT, 2 mM biotin) for 10 minutes and were subjected to NuPAGE 4%-12% gel (Invitrogen) and sliver staining (Pierce, 24612). The excised gel segments were subjected to mass spectrum analysis.

### Mass spectrum analysis
Following sliver staining, the gel segments were excised and subjected to in-gel trypsin digestion and dried. Ten microliters of peptides (dissolved with 0.1% formic acid) were auto-sampled onto a 100 μm × 10 cm fused silica emitter packed with reversed-phase ReproSil-Pur C18-AQ resin (3 μm and 120 Å; Ammerbuch, Germany). Linear gradients of 5–32% acetonitrile in 0.1% formic acid were used to elute the sample at a flow rate of 300 nl/min for 50 min. Mass spectrometry data were acquired with an LTQ Orbitrap Elite mass spectrometer (Thermo Fisher Scientific) equipped with a nanoelectrospray ion source (Proxeon Biosystems). Fragmentation in the LTQ was performed by collision-induced dissociation (normalized collision energy, 35%; activation Q, 0.250; activation time, 10 ms) with a target value of 3000 ions. In order to search the raw files, the SEQUEST engine against a database from the Uniprot protein sequence database was used.

### Cross-linking and Immunoprecipitation (CLIP) analysis
HEK293T cells were transfected with plasmids expressing mock or FLAG-tagged DDX3X and circRIG-I. After 6 hours of transfection, the culture medium was changed. Cells were then treated with 100 μM 4-thiouridine (abcam, ab143718) for 14 hours and subjected to UV-crosslinking on ice with 0.15 J/cm$^2$ at 365 nm wavelength. Subsequently, cells were collected and lysed with NP-40 lysis buffer (50 mM HEPES, 150 mM KCl, 2 mM EDTA, 0.5% NP-40, 0.5 mM DTT). After incubation with anti-FLAG M2 beads, the binding components were mixed with NT2 buffer (50 mM HEPES, 150 mM KCl, 1 mM MgCl$_2$, 0.05% NP-40) containing SDS, proteinase K (Roche, 3115879001) and RNase inhibitor at 56 °C. The RNA was then isolated by Trizol reagent and analyzed by real-time PCR.

### RNA pulldown
HEK293T cells expressing Mock and circRIG-I cells were lysed by RNA pulldown lysis buffer (20 mM Tris-HCl (pH = 7.5), 150 mM NaCl, 1 mM EDTA, 0.5% NP-40) with protease inhibitor and RNase inhibitor. Cell lysates were incubated with 6 μg of biotin-labeled probe at room temperature for 4 hours. Streptavidin agarose beads were washed by lysis buffer and blocked by yeast tRNA (500 ng/μL). Subsequently, whole cell lysates with streptavidin agarose beads were rotated for

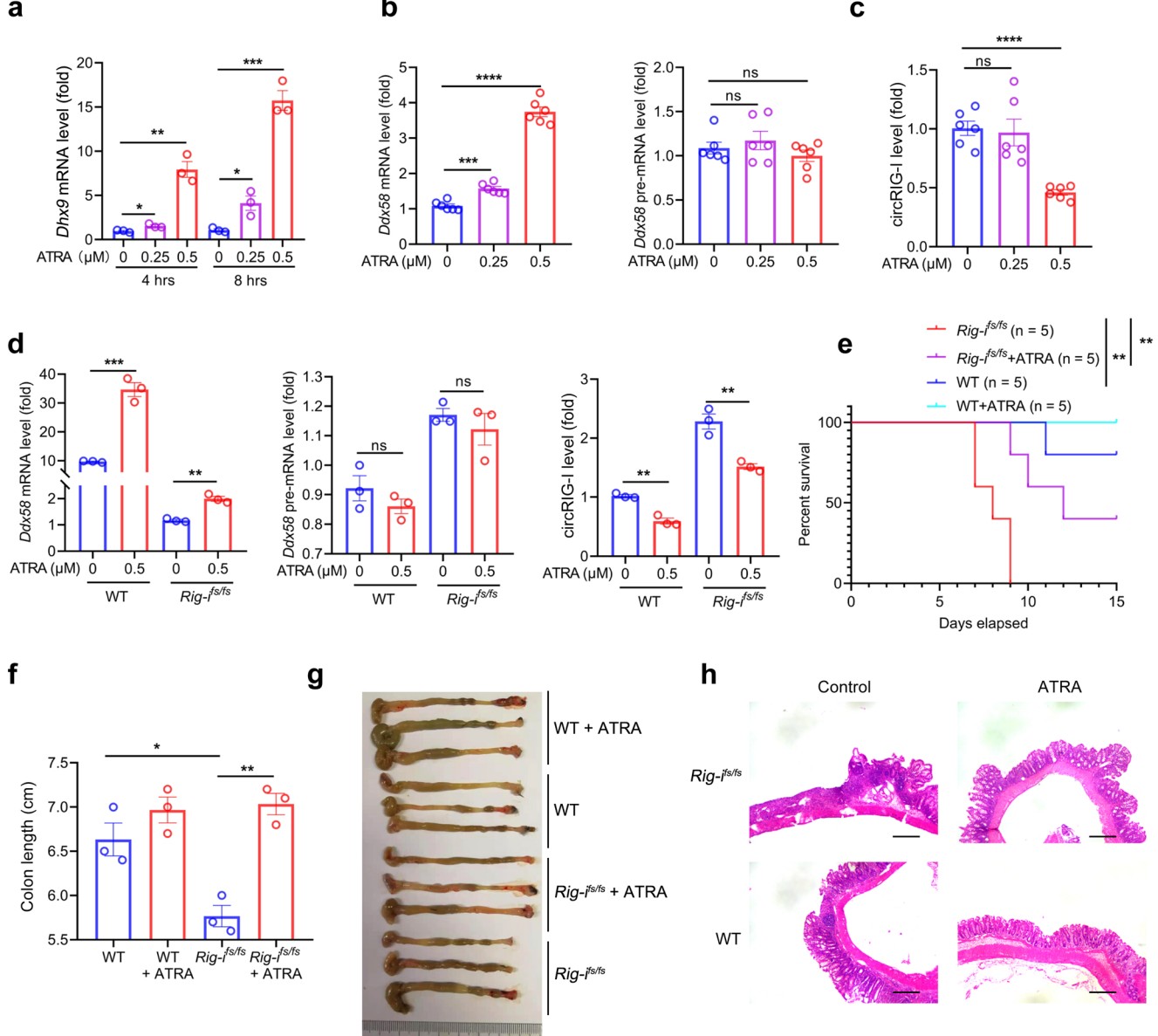

**Fig. 8 | All-trans retinoic acid alleviates colonic inflammatory lesion of *Rig-i*$^{fs/fs}$ mice. a** RT-qPCR analysis of *Dhx9* mRNA in MEFs with ATRA treatment (*n* = 6 cell cultures, mean ± s.e.m., ns, *$P$ = 0.0177 (4 h), *$P$ = 0.0204 (8 h), **$P$ = 0.0015, ***$P$ = 0.0002, two-tailed unpaired Student's *t*-test). **b-c** RT-qPCR analysis of *Ddx58* mRNA and pre-mRNA as well as circRIG-I expression in MEFs with ATRA treatment (*n* = 6 cell cultures, mean ± s.e.m., ns, not significant ($P$ > 0.05), ***$P$ = 0.0001, ****$P$ < 0.0001, two-tailed unpaired Student's *t*-test). **d** RT-qPCR analysis of mRNA and pre-mRNA of *Ddx58*, circRIG-I expression in wild-type (WT) and *Rig-i*$^{fs/fs}$ MEFs with ATRA treatment (*n* = 3 cell cultures, mean ± s.e.m., ns, not significant ($P$ > 0.05), **$P$ = 0.0012 (*Ddx58*), **$P$ = 0.0016 (WT, circRIG-I), **$P$ = 0.0051 (*Rig-i*$^{fs/fs}$, circRIG-I), ***$P$ = 0.0005, two-tailed unpaired Student's *t*-test). **e** Survival analysis of wild-type

(WT) and *Rig-i*$^{fs/fs}$ mice with or without ATRA treatment in DSS model (*n* = 5 mice, **$P$ = 0.0079 (*Rig-i*$^{fs/fs}$ vs. *Rig-i*$^{fs/fs}$ + ATRA), **$P$ = 0.0025 (WT vs. *Rig-i*$^{fs/fs}$), Log-rank (Mantel-Cox) test). **f-g** Colon length (**f**) and macroscopic evaluation (**g**) of wild-type (WT) and *Rig-i*$^{fs/fs}$ mice with or without ATRA treatment in DSS model (*n* = 3 mice, mean ± s.e.m., *$P$ = 0.0173, **$P$ = 0.0017, two-tailed unpaired Student's *t*-test). **h** Representative H&E staining pictures of colon tissues from wild-type (WT) and *Rig-i*$^{fs/fs}$ mice with or without ATRA treatment in DSS model. The scale bars represent 1000 μm. Data are collected from 2 independent experiments. The primers used for quantitative real-time PCR have been deposited in Supplementary Data 5. Source data are provided as a Source Data file.

1 hour. The binding components were washed by lysis buffer for three times and subjected to SDS-PAGE.

**In vitro RNA pulldown**

The biotin-labeled WT or framshift mutation of murine *Rig-i* RNA including exon 3 and intron 3 was obtained by T7 RNA polymerase kit (Promega, P2075) and Biotin RNA Labeling Mix (Roche, 11685597910). WT *Rig-i* RNA antisense sequence was used as control. Labeled RNA was incubated with cell lysates from HEK293T cells overexpressing DHX9-FLAG and RNA binding proteins were isolated by streptavidin

agarose beads (GE, 17-5113-01). The binding components were eluted by biotin (T1116), followed by SDS-Page.

**Statistics and reproducibility**

Statistical analysis was performed using Prism GraphPad software v7.0. Differences between two groups were calculated using a two-tailed Student's *t* test. Mice survival status was analyzed by Log-rank (Mantel-Cox) test. $P$ < 0.05 was considered significant. All experiments were independently replicated at least two times and similar results were generated.

**Reporting summary**

Further information on research design is available in the Nature Portfolio Reporting Summary linked to this article.

## Data availability

The RNA-seq data generated in this study have been deposited in the SRA database under accession codes PRJNA741380 and PRJNA759839. The circRNA-seq data generated in this study have been deposited in the SRA database under accession code PRJNA782066. The mass spectrometry proteomics data have been deposited to the ProteomeXchange Consortium via the PRIDE partner repository with the dataset identifier PXD037928 and PXD037929. The WES data generated in this study have been deposited in the SRA database under accession code PRJNA897090. Source data are provided with this paper. The remaining data are available within the Article, Supplementary Information or Source Data file.

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

## Acknowledgements

This work was supported by grants including the National Natural Science Foundation of China (Grants 82221003, 82022032, 81991505 and 82171826 to D.L.), Clinical Medicine Plus X-Young Scholars Project, Peking University (No. PKU2021LCXQ026 to D. L.), the Fundamental Research Funds for the Central Universities (No. BMU2018YJO03 to D. L) and Innovation Fund for Outstanding Doctoral Candidates of Peking University Health Science Center (No. BMU2021BSS001 to J.S.).

## Author contributions

J.S. and D.L. conceived and designed the experiments; J.S. and W.Z. performed most of the experiments and analyzed the data; W.T., L.M., Y.C., and Xuehui.Z. assisted in some experiments; Xuyang.Z. and Xin.Z. performed mass spectrometry analysis; D.L., X.D. and J.S. wrote the paper.

## Competing interests

The authors declare no competing interests.
