## [Peer Review File · Nature Communications]

Mutant RIG-I enhances cancer-related inflammation through activation of circRIG-I signalingReviewers' comments:

Reviewer #1 (Remarks to the Author): with expertise in RIG-1, anti-viral and cancer immunity

In their manuscript "Mutant RIG-I enhances cancer-related inflammation through activation of circRIG-I signaling" Song et al. report that they found heterozygous frame-shift mutations at position 359 that creates a premature stop codon. If the RNA were to be translated it would generate a protein that only contains the first 135 AA of RIG I and ends somewhere in the second CARD domain of RIG-I (the predicted molecular weight would be about 19 KD).

They then generate a mouse model that recapitulates this mutation in the murine RIG-I and show that this leads to a phenotype of enhanced interferon response to viruses and enhanced inflammatory response to DSS induced colitis and consequently presumably inflammation-triggered adenocarcinomas in the colon. Even though they analyze homozygous mice in these experiments – in contrast to the heterozygous patients.

So far the phenotype would fit to a gain of function mutation in RIG-I however they claim that there is no functional protein expressed and the mature mRNA of RIG-I expressed in only low amounts. In contrast they claim that the frame-shift mutation leads to a change in the secondary structure of the pre-mRNA of RIG-I, that this change reduces the interaction of the pre-mRNA of RIG-I with DHX9 and that this detachment in a not further elucidated way leads to an increase in a circular RNA derived from the pre-mRNA of RIG-I that they call CIRC-RIG-I. Overexpression of this circular RNA induces an interferon-response and the search of binding proteins of CIRC-RIG-I identifies DDXH3 as one possible binding protein. By showing that DDXH3 interacts in the presence of CIRC-RNA with MAVS, TBK1 and TRAFs they suggest that circ-RNA induces an Interferon-response via a DDXH3-MAVS-dependent way.

While I think the data showing/describing the frameshift mutation and its phenotype in the generated mouse model is valid and interesting - this part is not completely new. The really surprising and potentially exciting part of the paper, however is the proposed mechanism how a point mutation in RIG-I could lead to an enhanced interferon/inflammatory response via a circular RNA derived from the pre-mRNA.

Here however I have some mayor doubts about the mechanism and open questions remain to me:
- A potentially much easier explanation for the phenotype would be if the frameshift mutation leads to a translated protein with a gain of function phenotype. Already the first detection of RIG-I as an interferon inducing antiviral protein was based on a truncated cDNA clone only containing the first 285 AS (Yoneyama 2004). So far the shown evidence that the frameshift mutation RNA is not translated is rather weak and based on a western blot using an antibody that - though described to have its epitope in the N-terminal region - was generated by an antigen much larger than the predicted 135 AA long protein of the frame-shift mutation. The frame-shift mutation protein may therefore not been detected by the antibody.

The 135 AAs of the frame-shift protein would contain the complete first CARD of RIG-I and part of the second CARD. It is well known that constructs of the two CARDS of RIG-I are constitutively active and trigger MAVS-dependent signaling. A constitutive active RIG-I basically explains all of the phenotypic findings in RIG-I-fs/fs cells and animals. I therefore think a lot more evidence would be needed to proof that the phenotype is not based on the translated RIG-I FS/FS protein. Experiments in this direction would be testing the expression of RIG-I FS/FS constructs with N-terminal TAGs that can be detected by Tag-specific antibodies.

The PCR-assay detecting RIG-I RNA is based on primers binding to regions that are not translated any more and towards the 3'prime end of the cDNA. These parts of the RNA might be degraded by nonsense-decay mechanisms starting on the 3-prime end and still leave the 5-prime parts intact long enough to be translated into a protein.

To really show the existence of the relevant translated part of the RNA PCR-assays based on this region and northern blots are needed to assess if the transcripts do exist or not.

Even though not tested and discussed in the manuscript the concept of the circ-RIG-I RNA triggering a MAVS-dependent interferon-response in the context of a loss of function RIG-I that is not translated, to me, implies that either this circular RNA is by itself detected via MDA-5 or it must enhance an

MDA-5-dependent signal.

Maybe a revealing experiment would be here to cross the RIG-I fs/fs mice to MAVS -/- and MDA5 -/- mice. The prediction being that the phenotype should be rescued by the MAVS KO but still be partly there in the MDA-5 KO if the effect is mediated via a gain of function RIG-I protein and gone if it is mediated via an enhanced MDA5 signal in the context of a not translated RIG-I

- I have some difficulties to understand why in the context of this story the circ. RIG-I RNA should be so special that it is able to explain the phenotype. The circular RNA by itself is not changed or mutated by the FS-mutation. "only" a quantitative change in the amount of the circular RNA is also induced by the FS-mutation. However any interferon-signal that triggers the transcription of the "ISG" RIG-I induces a much higher amount of this circular RIG-I RNA but does not trigger the autoimmune-phenotype. And why do you not think the phenotype is triggered by one of the other of the 10 highest induced circular RNAs?

- I do not find the data of the extended figure 8 and figure 5a convincing that try to specifically knock down only the circular RIG-I-RNA. To me it seems these RNAs also knock-down the linear RIG-I RNA at least to some extent. Consequently, the result could also be explained by the knock-down of a translated part of the RIG-I RNA. What does happen if you use a shRNA that targets the linear RNA, do you see a similar phenotype?

- Is it really the MDA-5 signaling the pathway enhanced via CIRC-RIG I and DDX3X. Can you block the signal by knock-down of MDA5 or MAVS?

Reviewer #2 (Remarks to the Author): with expertise in CRC, circRNAs, immunity

Song et al., described "Mutant RIG-I enhances cancer-related inflammation through activation of circRIG-I signaling". Using sequencing analysis of blood samples from colon cancer patients, authors identified frameshift variants in Rig1 gene. They constructed Rig1-mutant mouse model, and found that Rig1 mutation activated type I IFN pathway in non-immune cells and contributed to DSS/AOM-induced colon cancer. Interestingly, Rig1 mutation abrogated RIG-1 protein expression and led to generation of circRIG1. Rig1 mutation impaired the interaction between Rig1 pre-mRNA and DHX9 which inhibits circRIG1 formation. CircRIG1 then activated DDX3X/MAVS/TRAF5/TBK1 pathway and increase innate immune response. Moreover, ATRA treatment ameliorated DSS-induced colitis by suppression of circRIG1 expression. This manuscript is very interesting. It aims to delineate how RIG1 mutation affects the development of colitis and colon cancer. However, the manuscript suffers from several weaknesses:

1. Fig 1l-n: Rig1 mutation also led to decrease of T cells in spleen, PP and mLN. How to explain this phenotype. Rig1 mutant mice were more susceptibility to colon cancer after DSS/AOM treatment. Rig1 is a very critical immune protein that regulates anti-tumor response. Thus, the change of immune cells in tumor tissues of DSS/AOM model should be measured.
2. Fig 2e: authors concluded that Rig1 mutation exacerbated inflammatory damage in an epithelial cell-intrinsic manner. To fully exclude the effect on immune cells, BMT assay using WT or Rig1fs/fs bone marrow cells as donor should be conducted.
3. Fig 2n, o: The upregulation of Isg15 and Tnf mRNAs was not obvious. After DSS or LPS treatment, TNF- α protein level could be tested by ELISA.
4. Fig. 4i: a pre-mRNA can produce many circRNA variants. After RIG1 mutation, how did other circRNA variants derived from RIG1 change?
5. authors demonstrated that circRIG1 interacted with DDX3X. To prove that circRIG1 exerts functions through DDX3X, authors should perform more experiments. At least, DDX3X should be knocked down in Rig1-mutant cells and then activation of type I IFN signaling should be evaluated after virus infection or LPS treatment.

Minor comments:

A typo in the abstract: "Though interaction with DDX3X" to "Through ..."

Reviewer #3 (Remarks to the Author): with expertise in circRNAs, cancer

The paper entitled Mutant RIG-I enhances cancer-related inflammation through activation of circRIG-I signaling, by Song et al. describes frameshift germline mutations in DDX58/RIG-I in patients with colon cancer and continue to study these mutations in CRISPR/Cas9 engineered mouse models. They found that the double mutant mice had an increased susceptibility to experimental colitis and colitis-associated colon cancer (CAC) and find that the mutation leads to generation of a circular RNA from the RIG-I pre-mRNA transcript. This circular RNA was shown to interact with DDX3X, thereby stimulating the MAVS/TRAF5/TBK1 signaling cascade, eventually causing an aberrant inflammatory response. The study is well-designed, and authors have performed a comprehensive series of experiments. The data are well-presented, and the paper is well-written. I believe that the authors provide sufficient evidence to support most of their conclusions. However, some points must be addressed before publication to truly understand its potential impact:

Major revisions

1. The colon cancer patients studied here, are very poorly described. Essentially no information is presented regarding age, gender, subtype, location, perineural invasion, lymphatic or vascular invasion, tumor differentiation, mismatch repair status, stage, treatment etc. While all these data are of importance, it is particularly important for this study to know if the patients carrying the frameshift germline mutations in DDX58/RIG-I had colitis-associated colon cancer.
2. There is no data presented to show if the circular RNA identified in the mouse model is conserved to human. Accordingly, it was also not shown if colon cancer patients with DDX58/RIG-I frameshift mutations have elevated levels of circRIG-I. These aspects are of key importance to link the findings from the mouse model to the discovery of the germline DDX58/RIG-I frameshift mutations in patients with colon cancer.
3. For the overexpression experiments of circRIG-I, the authors should do a Northern blotting +/- RNase R treatment to provide information about whether the vector produces concatemers and linear co-products. Also, the overexpression seems to be very efficient. How do the levels compare to endogenous levels? The authors should comment on whether the levels can be considered physiologically relevant.
4. Did the authors BLAST the shRNA sequences in a search for potential off-target effects?

Minor revisions

1. Regarding the Knock down experiments of circRIG-I, some of the used shRNAs are quite skewed on the back-splicing junction (indicated by |): (1#: CTCGGAGGTTTGAAG|GTTTCA; 2#: GAGGTTTGAAG|GTTTCAAAG; 3#: GTTTGAAG|GTTTCAAAGGGC; 4#: GAAG|GTTTCAAAGGGCTGAA). It was probably expected based on this that #1 and #4 knocked down the linear counterpart. The authors might want to comment on this.
2. The authors might want to comment on that frameshift germline mutations in DDX58/RIG-I have been observed before according to the COSMIC database (https://cancer.sanger.ac.uk/cosmic/gene/analysis?all_data=&coords=AA%3AAA&dr=&end=926&gd=&id=287669&ln=DDX58&seqlen=926&sn=large_intestine&start=1#ts). These data seem to be in accordance with the present findings and this could, for instance, be mentioned in the discussion.
3. The authors show that ATRA betters the cancer-related inflammation by suppression of circRIG-I expression, through promotion of DHX9. Since DHX9 is involved in the splicing and alternative splicing of many genes, could it be speculated that it has other effects and/or side-effects. They authors might want to comment on this in the discussion section.

4. Throughout there are sentences that have grammatical errors, please read the entire manuscript to carefully to address this.

Reviewer #4 (Remarks to the Author): with expertise in colitis, colorectal cancer

In the manuscript "Mutant RIG-I enhances cancer related inflammation through activation of circRIG-I signaling" Song et al. report on numerous findings based on a RIG-I variant.

The manuscript contains numerous experiments of different fields of innate signaling (cancer, inflammation, viral response). For me it is not clear why the authors want to connect this RIG-I variant to cancer? There is very limited data on cancer-related aspects in this manuscript. Detecting 3 out of 350 sporadic CRC patients with a variant is certainly too weak to make this claim. In addition using a very weak animal model for CAC is not sufficient per se and in particular that the effect of the RIG-1 variant could only be demonstrated on the inflammatory response. Did the authors find the variant in ulcerative colitis patients.

The authors do not investigate how the variant is distributed among the CMS types of CRC.

Whereas the experiments are well done the manuscript lacks focus completely. I cannot see that the tumor microenvironment of colon cancer is altered based on the variant. I would suggest to restructure the complete manuscript and focus on a cancer-independent inflammatory response of substantiate the cancer information in the manuscript.

The following is a point-by-point response to the reviewers' comments and questions.

Reviewers' comments:

Reviewer #1 (Remarks to the Author): with expertise in RIG-1, anti-viral and cancer immunity

In their manuscript "Mutant RIG-I enhances cancer-related inflammation through activation of circRIG-I signaling" Song et al. report that they found heterozygous frame-shift mutations at position 359 that creates a premature stop codon. If the RNA were to be translated it would generate a protein that only contains the first 135 AA of RIG I and ends somewhere in the second CARD domain of RIG-I (the predicted molecular weight would be about 19 KD).

They then generate a mouse model that recapitulates this mutation in the murine RIG-I and show that this leads to a phenotype of enhanced interferon response to viruses and enhanced inflammatory response to DSS induced colitis and consequently presumably inflammation-triggered adenocarcinomas in the colon. Even though they analyze homozygous mice in these experiments – in contrast to the heterozygous patients.

So far the phenotype would fit to a gain of function mutation in RIG-I however they claim that there is no functional protein expressed and the mature mRNA of RIG-I expressed in only low amounts.

In contrast they claim that the frame-shift mutation leads to a change in the secondary structure of the pre-mRNA of RIG-I, that this change reduces the interaction of the pre-mRNA of RIG-I with DHX9 and that this detachment in a not further elucidated way leads to an increase in a circular RNA derived from the pre-mRNA of RIG-I that they call CIRC-RIG-I. Overexpression of this circular RNA induces an interferon-response and the search of binding proteins of CIRC-RIG-I identifies DDXH3 as one possible binding protein. By showing that DDXH3 interacts in the presence of CIRC-RNA with MAVS, TBK1 and TRAFs they suggest that circ-RNA induces an Interferon-response via a DDXH3-MAVS-dependent way.

While I think the data showing/describing the frameshift mutation and its phenotype in the generated mouse model is valid and interesting - this part is not completely new. The really surprising and potentially exciting part of the paper, however is the proposed mechanism how a point mutation in RIG-I could lead to an enhanced interferon/inflammatory response via a circular RNA derived from the pre-mRNA.

[Response] We really appreciate the reviewer's professional and constructive comments. We have addressed each comment, and details are listed as follows.

Here however I have some mayor doubts about the mechanism and open questions

remain to me:

- A potentially much easier explanation for the phenotype would be if the frameshift mutation leads to a translated protein with a gain of function phenotype. Already the first detection of RIG-I as an interferon inducing antiviral protein was based on a truncated cDNA clone only containing the first 285 AS (Yoneyama 2004). So far the shown evidence that the frameshift mutation RNA is not translated is rather weak and based on a western blot using an antibody that - though described to have its epitope in the N-terminal region - was generated by an antigen much larger than the predicted 135 AA long protein of the frame-shift mutation. The frame-shift mutation protein may therefore not been detected by the antibody.

The 135 AAs of the frame-shift protein would contain the complete first CARD of RIG-I and part of the second CARD. It is well known that constructs of the two CARDS of RIG-I are constitutively active and trigger MAVS-dependent signaling. A constitutive active RIG-I basically explains all of the phenotypic findings in *Rig-I*-fs/fs cells and animals. I therefore think a lot more evidence would be needed to proof that the phenotype is not based on the translated RIG-I FS/FS protein. Experiments in this direction would be testing the expression of RIG-I FS/FS constructs with n-terminal TAGs that can be detected by Tag-specific antibodies.

[Response] According to these suggestions, the following experiments were carried out to determine whether the *RIG-I* gene with frameshift mutation can be translated into protein or not. We first cloned the wild-type *RIG-I*, the frameshift mutant *RIG-I* (RIG-I₁₋₁₄₀) or the active *RIG-I* truncation (RIG-I₁₋₂₈₄) into pEGFP-C1 vector, respectively. As shown in **Response Figure 1a**, the expression of RIG-I₁₋₁₄₀ was detected by both anti-GFP antibody and anti-RIG-I antibody, which further confirmed the reliability of anti-RIG-I antibody. Considering the fact that the expression level of RIG-I₁₋₁₄₀ was lower than wild-type RIG-I or RIG-I₁₋₂₈₄, the frameshift mutant protein may be undetectable by the antibody. We therefore employed mass spectrum assay to determine whether the *RIG-I* gene with frameshift mutation can be translated into protein in *Rig-i*^{fs/fs} MEFs. Despite many peptides covering the full length of RIG-I were identified in wild-type MEFs, no peptide was detected in *Rig-i*^{fs/fs} MEFs, which is consistent with the western blot assay (**Response Figure 1b**). Our data thus support the notion that frameshift mutation of *Rig-i* impairs its mRNA translation into protein.

Even though little protein was translated, we still want to assess the role of RIG-I₁₋₁₄₀ in modulation of host innate immunity. Unlike the wild-type RIG-I and the active RIG-I truncation (RIG-I₁₋₂₈₄), RIG-I₁₋₁₄₀ exerted little effects on host antiviral immune response (**Response Fig. 1c**). In light of the RIG-I₁₋₁₄₀ containing the complete the first CARD and part of the second CARD domain of RIG-I, we then used co-immunoprecipitation assay to study whether RIG-I₁₋₁₄₀ can interact with MAVS. As shown in **Response Figure 1d**, both the active RIG-I truncation (RIG-I₁₋₂₈₄) and RIG-I₁₋₁₄₀ interacted with MAVS. To compare the signaling transduction capacity between RIG-I₁₋₂₈₄ and RIG-I₁₋₁₄₀, we used the mitochondrial and cytoplasmic extraction assay. Compared with the predominant accumulation of RIG-I₁₋₂₈₄ on mitochondria, RIG-I₁₋₁₄₀ was largely localized in cytoplasm (**Response Fig. 1e**). Moreover, when RIG-I₁₋₁₄₀ were co-transfected with RIG-I₁₋₂₈₄ into HEK293T cells, the localization of RIG-I₁₋₂₈₄

on mitochondria were attenuated and localized mainly in the cytoplasm (**Response Fig. 1e**). Ensued confocal assay further confirmed this result (**Response Fig. 1f**). Since the dimerization of RIG-I is required for its-mediated type-I interferon signaling activation, we thus hypothesized that RIG-I₁₋₁₄₀ was involved in this process. To this end, we co-transfected vector encoding GFP-RIG-I₁₋₂₈₄ with empty vector, vector encoding FLAG-RIG-I₁₋₂₈₄ or vector encoding FLAG-RIG-I₁₋₁₄₀ into HEK293T cells. As expected, RIG-I₁₋₁₄₀ also interacted with RIG-I₁₋₂₈₄ (**Response Figure 1g**). Moreover, presence of RIG-I₁₋₁₄₀ impaired the association of RIG-I₁₋₂₈₄ with MAVS (**Response Figure 1h**), consequently restricting the transduction of innate immune signaling.

Collectively, our data demonstrated that frameshift mutation of *Rig-i* disrupts its mRNA translation into protein and that even if translated, the function of RIG-I₁₋₁₄₀ can hardly activate host innate immune response, which conflicts with the phenotype of *Rig-i*^{fs/fs} mice.

Response Figure 1. Structural incompleteness of N-terminal domain of RIG-I impairs its biological function.

(a) Immunoblot analysis of the expression of RIG-I or its truncations including RIG-I₁₋₁₄₀ and RIG-I₁₋₂₈₄ in HEK293T cells transfected with indicated vectors with GFP tag detected by anti-GFP antibody and anti-RIG-I antibody, respectively.

(b) Mass spectrum analysis of RIG-I expression in wild-type and *Rig-1^{fs/fs}* MEFs.

Protein peptides detected in wild-type MEFs were shown.

(c) RT-qPCR analysis of the RNA level of SeV in HEK293T cells transfected with indicated vectors following SeV infection (n = 3 biological replicates, mean \pm s.e.m.).

(d) HEK293T cells were co-transfected with vectors encoding MAVS-FLAG and RIG-I₁₋₂₈₄-GFP or RIG-I₁₋₁₄₀-GFP. At 24 hours later, cell lysates were immunoprecipitated with anti-FLAG antibody and protein A/G agarose beads and analyzed by immunoblot with anti-GFP antibody.

(e) Subcellular localization of RIG-I₁₋₂₈₄ and MAVS in HEK293T cells transfected with or without RIG-I₁₋₁₄₀ analyzed by mitochondria-cytoplasm extraction.

(f) Confocal examination of the localization of RIG-I₁₋₂₈₄ in HEK293T cells in presence or absence of RIG-I₁₋₁₄₀. The scale bars represent 20 μ m.

(g) HEK293T cells were co-transfected with vectors encoding RIG-I₁₋₂₈₄-GFP and RIG-I₁₋₂₈₄-FLAG or RIG-I₁₋₁₄₀-FLAG. At 24 hours later, cell lysates were immunoprecipitated with anti-FLAG antibody and protein A/G agarose beads and analyzed by immunoblot with anti-GFP antibody.

(h) HEK293T cells were co-transfected with vectors encoding MAVS-FLAG and RIG-I₁₋₂₈₄-GFP in presence or absence of RIG-I₁₋₁₄₀-GFP. At 24 hours later, cell lysates were immunoprecipitated with anti-FLAG antibody and protein A/G agarose beads and analyzed by immunoblot with anti-GFP antibody.

The **Response Figure 1b** was also shown in **Extended Data Fig. 4** in revised version.

The PCR-assay detecting RIG-I RNA is based on primers binding to regions that are not translated any more and towards the 3'prime end of the cDNA. These parts of the RNA might be degraded by nonsense-decay mechanisms starting on the 3-prime end and still leave the 5-prime parts intact long enough to be translated into a protein.

To really show the existence of the relevant translated part of the RNA PCR-assays based on this region and northern blots are needed to asses if the transcripts do exist or not.

[Response] As suggested, we designed two pairs of primers targeting the coding region (CDS) of *Rig-i* gene and repeated the experiments in **Figure 4a-4c**. As shown in **Response Figure 2a and 2b**, the transcription of *Rig-i* was also downregulated in *Rig-i^{fs/fs}* MEFs with or without virus infection both *in vitro* and *in vivo*. Moreover, we used probes that hybridize with the splicing junction to distinguish circRIG-I and probes that hybridize with exon 5 to distinguish circRIG-I and its host gene, RIG-I, by northern blotting (**Response Fig. 2c**), which further confirmed that existence of circRIG-I.

Response Figure 2. Assessment of status of circRIG-I and its linear counterpart in *Rig-1^{fs/fs}* cells.

(a-b) RT-qPCR analysis of the mRNA level of *Ddx58* in wild-type (WT) and *Rig-1^{fs/fs}* MEFs with viral infection by the primers targeting the coding region (CDS) of *Rig-1* gene (Left). (n = 3 biological replicates, mean \pm s.e.m., **** P < 0.0001, two-tailed unpaired Student's t-test). RT-qPCR analysis of the mRNA level of *Ddx58* in liver and spleen from wild-type (WT) and *Rig-1^{fs/fs}* mice followed infection with VSV (Right) (n = 4 biological replicates, mean \pm s.e.m., **** P < 0.0001, two-tailed unpaired Student's t-test). Primers used to detect *Ddx58* mRNA level were as follows: Forward, 5-AAGAGCCAGAGTGTTCAGAATCT-3; Reverse, 5-AGCTCCAGTTGGTAATTTCTTGG-3 (a). Forward, 5-AATATGTGCCCTACTGGTTGT-3; Reverse, 5-CGAAGAAGACCACTTTCCTTT-3 (b).

(c) Northern blotting analysis of circRIG-I and RIG-I mRNA levels by hybridization with exon 5 (upper) and exon 5-exon 12 junction (lower) probes with and without RNase R treatment. ACTB mRNA with or without RNase R treatment was detected as a control.

The Response Figure 2c was also shown in Fig. 4k in revised version.

Even though not tested and discussed in the manuscript the concept of the cir-RIG-I RNA triggering a MAVS-dependent interferon-response in the context of a loss of function RIG-I that is not translated, to me, implies that either this circular RNA is by itself detected via MDA-5 or it must enhances an MDA-5-dependent signal.

*Maybe a revealing experiment would be here to cross the RIG-I *fs/fs* mice to MAVS *-/-**

and MDA5 $-/-$ mice. The prediction being that the phenotype should be rescued by the MAVS KO but still be partly there in the MDA-5 KO if the effect is mediated via a gain of function RIG-I protein and gone if it is mediated via an enhanced MDA5 signal in the context of a not translated RIG-I.

[Response] According to these suggestions, the following experiments were carried out to determine whether MDA5 is involved in the circRIG-I-mediated innate immune signaling. We first performed RNA pulldown to assess the association between circRIG-I and MDA5. As shown in **Response Fig. 3a**, circRIG-I has no physical interaction with MDA5. We then used shRNA to knockdown of the endogenous MDA5, MAVS, DDX3X or IRF3 in both wild-type and *Rig-i^{fs/fs}* MEFs, respectively (**Response Fig. 3b**). Following infection with VSV-GFP, although loss of MAVS partly impaired the antiviral effects of *Rig-i^{fs/fs}* MEFs, the lower percentage of GFP⁺ *Rig-i^{fs/fs}* MEFs was observed when endogenous MDA5 was silenced as compared with wild-type MEFs (**Response Fig. 3c**). Moreover, identical high viral titration and low level of ISGs were detected in both wild-type and *Rig-i^{fs/fs}* MEFs when endogenous DDX3X or IRF3 were knocked down (**Response Fig. 3c and 3d**). Our data thus demonstrate that DDX3X-IRF3 signaling is essential for the antiviral effects of *Rig-i^{fs/fs}* MEFs.

To further confirm the result that MDA5 is not involved in this process, we used the Herpes simplex virus (HSV), a kind of DNA virus, to infect the wild-type and *Rig-i^{fs/fs}* MEFs. Unlike RNA virus, DNA virus can be recognized by AIM2 or cGAS rather than MDA5 or RIG-I, which subsequently activates host innate immune signaling. As shown in **Response Figure 4a**, similar to the results using RNA virus, lower viral titers and higher expression of ISGs were detected in *Rig-i^{fs/fs}* MEFs compared with wild-type MEFs. Consistent with these *in vitro* results, the limited titration of HSV and enhanced expression of ISGs were also observed in livers and lungs from *Rig-i^{fs/fs}* mice as compared with those from wild-type mice (**Response Fig. 4b and 4c**). Our data thus suggest that frameshift mutation of *Rig-i* can stimulate host innate immune response in an MDA5-independent manner.

Collectively, our data suggest that MDA5 is not involved in the signaling activated by frameshift mutation of *Rig-i*.

Response Figure 3. MDA5 is not involved in the signaling of frameshift mutation of *Rig-i*.

(a) Vectors encoding DDX3X-FLAG or MDA5-FLAG as well as circRIG-I were co-transfected into HEK293T cells. After 24 hours of transfection, RNA pulldown assay was performed by biotin labeled nucleotide probe specific to circRIG-I.

(b) Immunoblot confirmation of the effectiveness of shRNA against *MDA5*, *MAVS*,

IRF3 as well as *DDX3X* in wild-type and *Rig-^{fs/fs}* MEFs, respectively.

(c) Flow cytometric analysis of GFP⁺ wild-type (WT) and *Rig-^{fs/fs}* MEFs when endogenous MDA5, MAVS, IRF3 and DDX3X were knocked down with the treatment of VSV-GFP, respectively (n = 3 biological replicates, mean ± s.e.m., ns, not significant ($P > 0.05$), *** $P < 0.005$, **** $P < 0.0001$, two-tailed unpaired Student's t-test).

(d) RT-qPCR analysis of the mRNA levels of *Ifnβ* and other ISGs in wild-type (WT) and *Rig-^{fs/fs}* MEFs when endogenous MDA5, MAVS, IRF3 and DDX3X were knocked down following VSV infection (n = 4 biological replicates, mean ± s.e.m., ns, not significant ($P > 0.05$), ** $P < 0.01$, *** $P < 0.005$, **** $P < 0.0001$, two-tailed unpaired Student's t-test).

The **Response Figure 3b-d** were also shown in **Extended Data Fig. 13a-c** in the revised version.

Response Figure 4. Frameshift mutation of *Rig-i* enhances host immune response against HSV infection.

(a) RT-qPCR analysis of HSV as well as the mRNA level of *Ifnβ* and other ISGs in wild-type and *Rig-i^{fs/fs}* MEFs with or without HSV infection (n = 3 biological replicates, mean ± s.e.m., **P* < 0.05, ***P* < 0.01, ****P* < 0.001, *****P* < 0.0001, two-tailed unpaired Student's t-test).

(b-c) RT-qPCR analysis of HSV as well as the mRNA level of *Ifnβ* and other ISGs in liver (b) and lung (c) from wild-type (WT) and *Rig-i^{fs/fs}* mice following HSV infection (n = 4 biological replicates, mean ± s.e.m., **P* < 0.05, ***P* < 0.01, ****P* < 0.001, two-tailed unpaired Student's t-test).

- I have some difficulties to understand why in the context of this story the circ. RIG-I RNA should be so special that it is able to explain the phenotype. The circular RNA by itself is not changed or mutated by the FS-mutation. "only" a quantitative change in the amount of the circular RNA is also induced by the FS-mutation. However any interferon-signal that triggers the transcription of the "ISG" RIG-I induces a much higher amount of this circular RIG-I RNA but does not trigger the autoimmune-phenotype. And why do you not think the phenotype is triggered by one of the other of the 10 highest induced circular RNAs?

[Response] We agree with the reviewer that the circRIG-I by itself is not changed or mutated by this frameshift mutation. However, our data revealed that this frameshift mutation impaired the stem loop structure of *RIG-I* RNA and blocked the binding of DHX9 on the *Rig-i* transcript that localized on chromosome 4. In light of the essential role of DHX9 in modulation of circRNA biogenesis, it is conceivable that the frameshift mutation of *Rig-i* can trigger circRNA generation from the *RIG-I* transcript. Although the other nine circular RNAs were also upregulated in *Rig-i^{fs/fs}* MEFs upon VSV infection, they were localized in other chromosomes rather than chromosome 4. We thus consume that the frameshift mutation initially triggers the induction of circRIG-I, which in turn exaggerates type-I interferon signaling, consequently resulting in the upregulation of other circular RNAs. Referring to the results of overexpression of circRIG-I in **Figure 5**, we thus assume that the upregulation of circRIG-I increases the susceptibility of mice to inflammation or autoimmune diseases. Indeed, considering the upregulation of the other nine circular RNAs during viral infection, we will continue our study to analyze their biological function in our future work.

- I do not find the data of the extended figure 8 and figure 5a convincing that try to specifically knock down only the circular RIG-I-RNA. To me it seems these RNAs also knock-down the linear RIG-I RNA at least to some extent. Consequently, the result could also be explained by the knock-down of a translated part of the RIG-I RNA. What does happen if you use a shRNA that targets the linear RNA, do you see a similar phenotype?

[Response] Because of the critical role of RIG-I in recognition of RNA virus, knockdown the expression of RIG-I can block the activation of host antiviral immune response. It is conceivable that knockdown the linear *RIG-I* mRNA will exhibit similar phenotype as the cells when circRIG-I was knocked down. To assess the role of circRIG-I rather than RIG-I, we thus used short hairpin RNA (shRNA) that targets the back-splice junction of circRIG-I to selectively knockdown circRIG-I without affecting its linear counterpart *Rig-i*. As shown in **Extended Data Fig. 8a**, unlike the downregulation of the linear *Rig-i* mRNA when pLKO-circRIG-I-1 or pLKO-circRIG-I-4 were stably transfected into MEFs, the expression of *Rig-i* mRNA was hardly affected in cell stably transfected with pLKO-circRIG-I-2 or pLKO-circRIG-I-3. Moreover, the level of circRIG-I was selectively downregulated in these cells. We thus employed pLKO-circRIG-I-2 and pLKO-circRIG-I-3 to analyze the role of circRIG-I

in modulation of host antiviral immunity.

- Is it really the MDA-5 signaling the pathway enhanced via CIRC-RIG I and DDX3X. Can you block the signal by knock-down of MDA5 or MAVS?

[Response] As suggested, we used shRNA to knockdown of the endogenous MDA5, MAVS, DDX3X or IRF3 in both wild-type and *Rig-^{fs/fs}* MEFs, respectively (**Response Fig. 5a**). Following infection with VSV-GFP, although loss of MAVS partly impaired the antiviral effects of *Rig-^{fs/fs}* MEFs, the lower percentage of GFP⁺ *Rig-^{fs/fs}* MEFs was observed when endogenous MDA5 was silenced as compared with wild-type MEFs (**Response Fig. 5b**). Moreover, identical high viral titration and low level of ISGs were detected in both wild-type and *Rig-^{fs/fs}* MEFs when endogenous DDX3X or IRF3 were knocked down (**Response Fig. 5b and 5c**). Our data thus demonstrate that, rather than MDA5, DDX3X-IRF3 signaling is essential for the antiviral effects of *Rig-^{fs/fs}* MEFs.

Response Figure 5. DDX3X and IRF3 are required for the antiviral effects of the frameshift mutation of *Rig-i*.

(a) Immunoblot confirmation of the effectiveness of shRNA against *MDA5*, *MAVS*, *IRF3* as well as *DDX3X* in both wild-type and *Rig-i^{fs/fs}* MEFs, respectively.

(b) Flow cytometric analysis of GFP⁺ wild-type (WT) and *Rig-i^{fs/fs}* MEFs when endogenous *MDA5*, *MAVS*, *IRF3* and *DDX3X* were knocked down with the treatment

of VSV-GFP, respectively (n = 3 biological replicates, mean \pm s.e.m., ns, not significant ($P > 0.05$), *** $P < 0.005$, **** $P < 0.0001$, two-tailed unpaired Student's t-test).

(c) RT-qPCR analysis of the mRNA levels of *Ifn β* and other ISGs in wild-type (WT) and *Rig-1^{fs/fs}* MEFs when endogenous *MDA5*, *MAVS*, *IRF3* and *DDX3X* were knocked down following VSV infection (n = 3 biological replicates, mean \pm s.e.m., ns, not significant ($P > 0.05$), ** $P < 0.01$, *** $P < 0.005$, **** $P < 0.0001$, two-tailed unpaired Student's t-test).

These data were also shown in **Extended Data Fig. 13** in revised version.

Reviewer #2 (Remarks to the Author): with expertise in CRC, circRNAs, immunity

Song et al., described "Mutant RIG-I enhances cancer-related inflammation through activation of circRIG-I signaling". Using sequencing analysis of blood samples from colon cancer patients, authors identified frameshift variants in Rig1 gene. They constructed Rig1-mutant mouse model, and found that Rig1 mutation activated type I FIN pathway in non-immune cells and contributed to DSS/AOM-induced colon cancer. Interestingly, Rig1 mutation abrogated RIG-1 protein expression and led to generation of circRIG1. Rig1 mutation impaired the interaction between Rig1 pre-mRNA and DHX9 which inhibits circRIG1 formation. CircRIG1 then activated DDX3X/MAVS/TRAF5/TBK1 pathway and increase innate immune response. Moreover, ATRA treatment ameliorated DSS-induced colitis by suppression of circRIG1 expression. This manuscript is very interesting. It aims to delineate how RIG1 mutation affects the development of colitis and colon cancer. However, the manuscript suffers from several weaknesses:

[Response] We would first like to express our sincere thanks to the reviewer for the positive comments and affirmation of our study. The reviewer's comments are very helpful and of value for improving the quality of our data and manuscript. Accordingly, we have addressed each of the comments as follows.

1. Fig 1l-n: Rig1 mutation also led to decrease of T cells in spleen, PP and mLN. How to explain this phenotype. Rig1 mutant mice were more susceptibility to colon cancer after DSS/AOM treatment. Rig1 is a very critical immune protein that regulates anti-tumor response. Thus, the change of immune cells in tumor tissues of DSS/AOM model should be measured.

[Response] Although the percentage of splenic T cells was decreased in spleen from *Rig-1^{fs/fs}* mice as compared with those in wild-type mice, the total amount of immune cells in spleen was increased *Rig-1^{fs/fs}* mice as relative to wild-type mice, indicating enhanced inflammatory response was triggered in *Rig-1^{fs/fs}* mice after DSS/AOM treatment. Unlike the beneficial effects of host anti-tumor immunity, the inflammation plays a detrimental role in colon carcinogenesis. To determine whether lymphocytes in peripheral lymphoid organ migrate to the inflammatory foci, we thus repeated the AOM-DSS experiment and isolated lamina propria mononuclear cells (LPMCs) from the tumor area. Through analysis by flow cytometry, we found that the percentage of T

cells were increased in LPMCs from *Rig-^{fs/fs}* mice as compared with wild-type mice (**Response Fig. 6**). Reciprocally, the percentage of B cells was reduced in LPMCs from *Rig-^{fs/fs}* mice as compared with wild-type mice (**Response Fig. 6**). Taken together, frameshift mutation of RIG-I augments inflammatory response in colon and facilitates colon cancer development.

Response Figure 6. Frameshift mutant *RIG-I* enhances inflammatory response in colon during tumor development.

Flow cytometric analysis of frequency of CD4⁺ T cell, CD8⁺ T cell and B cell subsets in lamina propria mononuclear cells (LPMCs) isolated from wild-type (WT) and *Rig-^{fs/fs}* mice treated with AOM/DSS (n = 5 mice, mean ± s.e.m., ****P < 0.0001, two-tailed unpaired Student’s t-test).

These data were also shown in **Fig. 1m** in revised version.

2. Fig 2e: authors concluded that Rig1 mutation exacerbated inflammatory damage in an epithelial cell-intrinsic manner. To fully exclude the effect on immune cells, BMT assay using WT or Rig1fs/fs bone marrow cells as donor should be conducted.

[Response] As suggested, in order to exclude the effect of *Rig-i* frameshift mutation on immune cells, we have performed bone marrow transplantation assay to reconstitute wild-type or *Rig-^{fs/fs}* immune system in wild-type mice. Transplanted mice were left for 30 days to allow for complete reconstitution of the immune system and then subjected to the DSS treatment. Following acute DSS colitis induction, the reconstituted mice with *Rig-^{fs/fs}* donor bone marrow exhibited similar susceptibility to colitis as the mice received with wild-type donor bone marrow did, as determined by weight loss, survival curve and gross tissue evaluation (**Response Fig. 7a-c**). Our data thus demonstrate that frameshift mutation of *Rig-i* in non-immune cell rather than immune cell contributes to the severe inflammatory damage.

Response Figure 7. *Rig-i* frameshift mutation exacerbates inflammatory damage in an immune cells-independent manner.

(a-c) Bone marrow from wild-type (WT) and *Rig-i^{fs/fs}* mice were transplanted into wild-type mice with irradiated (1000 cGy/mouse), respectively. 30 days after transplantation, mice were treated with DSS and assessed by body weight, survival time (n = 5 mice, ns, not significant ($P > 0.05$), Log-rank (Mantel-Cox) test) and colon length (n = 3 mice, mean \pm s.e.m., ns, not significant ($P > 0.05$), two-tailed unpaired Student's t-test).

These data were also shown in **Extended Data Fig. 7h-j** in revised version.

3. Fig 2n, o: The upregulation of *Isg15* and *Tnf* mRNAs was not obvious. After DSS or LPS treatment, TNF- α protein level could be tested by ELISA.

[Response] As suggested, we have used ELISA assay to detect the production of pro-inflammatory cytokines. Following acute DSS colitis induction, both serum and colon tissue interstitial fluid were harvested and measured by ELISA assay. As shown in **Response Figure 8a**, greater amounts of pro-inflammatory cytokines including IL1 β and TNF α were released in blood and extracellular matrix in *Rig-i^{fs/fs}* mice as compared with wild-type mice. Similar results were also observed in mice following LPS challenge (**Response Fig. 8b**).

Response Figure 8. Effects of frameshift mutation of *Rig-i* on the protein levels of IL1β and TNFα.

(a) The protein levels of IL1β and TNFα in serum or colonic interstitial fluid from wild-type and *Rig-i^{fs/fs}* mice after DSS treatment were assessed by ELISA assay (n = 4 mice, mean ± s.e.m., **P* < 0.05, ***P* < 0.01, two-tailed unpaired Student's t-test).

(b) The protein levels of IL1β and TNFα in serum or different tissue interstitial fluid (liver or lung) from wild-type and *Rig-i^{fs/fs}* mice after LPS treatment were assessed by ELISA assay (n = 4 mice, mean ± s.e.m., **P* < 0.05, ***P* < 0.01, ****P* < 0.001 two-tailed unpaired Student's t-test).

The **Response Figure 8a** was also shown in **Extended Data Fig. 7a,b** in revised version.

4. Fig. 4i: a pre-mRNA can produce many circRNA variants. After RIG1 mutation, how did other circRNA variants derived from RIG1 change?

[Response] To comprehensively analyze the circRNA variants derived from frameshift mutation of *RIG-I*, we treated ribosomal RNA-depleted total RNAs from virus-infected WT and *Rig-i^{fs/fs}* MEFs with RNase R to degrade linear RNAs and enrich circRNAs, eventually subjected to high-throughput RNA sequencing. We then screened key circRNAs according to the following criteria: i. Intensity of circRNA induction; ii. Chromosome localization (Reason: the frameshift mutation occurred in the *Rig-i* transcript that localized on chromosome 4); iii. Species conservativeness. Among circRNA variants, we identified the mmu-DDX58_0004 (circAtlas ID)

(<http://circatlas.biols.ac.cn/>), which was significantly upregulated in the transcript of *Rig-i* from *Rig-i^{fs/fs}* MEFs upon viral infection. The murine circRIG-I is conserved to human (<http://yang-laboratory.com/circpedia/search>), which was further confirmed by our experiments (**Response Fig. 9a and 9b**). Above all, our data support the notion that circRIG-I is critical for the stimulatory effects on inflammation by the frameshift mutation of *Rig-i*.

Response Figure 9. circRIG-I is conserved between human and mice.

(a) Human circRIG-I can be interrogated by the CIRCpedia V2 database (<http://yang-laboratory.com/circpedia/search>).

(b) Sequence of human circRIG-I junction site was shown in the colon cancer sample. The **Response Figure 9b** was also shown in **Fig. 5a** in revised version.

5. authors demonstrated that circRIG1 interacted with DDX3X. To prove that circRIG1 exerts functions through DDX3X, authors should perform more experiments. At least, DDX3X should be knocked down in Rig1-mutant cells and then activation of type I IFN signaling should be evaluated after virus infection or LPS treatment.

[**Response**] As suggested, we used shRNA to knockdown of the endogenous MDA5, MAVS, DDX3X or IRF3 in both wild-type and *Rig-i^{fs/fs}* MEFs, respectively (**Response Fig. 10A**). Following infection with VSV-GFP, although loss of MAVS partly impaired the antiviral effects of *Rig-i^{fs/fs}* MEFs, the lower percentage of GFP⁺ *Rig-i^{fs/fs}* MEFs was observed when endogenous MDA5 was silenced as compared with wild-type MEFs (**Response Fig. 10B and 10C**), indicating MDA5 is not involved in circRIG-I-mediated signaling pathway. Moreover, identical high viral titration and low levels of ISGs were detected between wild-type and *Rig-i^{fs/fs}* MEFs when endogenous DDX3X or IRF3 were knocked down (**Response Fig. 10B and 10C**). Our data thus demonstrate that DDX3X-IRF3 signaling is essential for the antiviral effects of *Rig-i^{fs/fs}* MEFs.

of VSV-GFP, respectively (n = 3 biological replicates, mean \pm s.e.m., ns, not significant ($P > 0.05$), *** $P < 0.005$, **** $P < 0.0001$, two-tailed unpaired Student's t-test).

(c) RT-qPCR analysis of the mRNA levels of *Ifn β* and other ISGs in wild-type (WT) and *Rig-1^{fs/fs}* MEFs when endogenous *MDA5*, *MAVS*, *IRF3* and *DDX3X* were knocked down following VSV infection (n = 4 biological replicates, mean \pm s.e.m., ns, not significant ($P > 0.05$), ** $P < 0.01$, *** $P < 0.005$, **** $P < 0.0001$, two-tailed unpaired Student's t-test).

These data were also shown in **Extended Data Fig. 13** in revised version.

Minor comments:

A typo in the abstract: "Though interaction with DDX3X" to "Through ..."

[Response] We are grateful to the reviewer for such careful review of our work, and we regret this error. We have replaced "Though interaction with DDX3X" with "Through" in the revised version.

Reviewer #3 (Remarks to the Author): with expertise in circRNAs, cancer

The paper entitled Mutant RIG-I enhances cancer-related inflammation through activation of circRIG-I signaling, by Song et al. describes frameshift germline mutations in DDX58/RIG-I in patients with colon cancer and continue to study these mutations in CRISPR/Cas9 engineered mouse models. They found that the double mutant mice had an increased susceptibility to experimental colitis and colitis-associated colon cancer (CAC) and find that the mutation leads to generation of a circular RNA from the Rig-i pre-mRNA transcript. This circular RNA was shown to interact with DDX3X, thereby stimulating the MAVS/TRAF5/TBK1 signaling cascade, eventually causing an aberrant inflammatory response. The study is well-designed, and authors have performed a comprehensive series of experiments. The data are well-presented, and the paper is well-written. I believe that the authors provide sufficient evidence to support most of their conclusions. However, some points must be addressed before publication to truly understand its potential impact:

[Response] We appreciate the reviewer's comments and instructions for improving our data and manuscript. We have done an additional series of experiments and addressed all the comments as below:

Major revisions

1. The colon cancer patients studied here, are very poorly described. Essentially no information is presented regarding age, gender, subtype, location, perineural invasion, lymphatic or vascular invasion, tumor differentiation, mismatch repair status, stage, treatment etc. While all these data are of importance, it is particularly important for this study to know if the patients carrying the frameshift germline mutations in DDX58/RIG-I had colitis-associated colon cancer.

[Response] As suggested, we have presented the clinical information in the **Extended**

Data Table 3. Through analysis of the clinical information from patients carrying the frameshift mutations, we found that all tumors were ulcerative adenocarcinoma of colon (pT4aN1a, pT3N0 and pT3aN1a) despite we did not find whether there was a history of colitis. In addition to the frameshift mutation, we also identified human circRIG-I that was conserved to mice (**Response Fig. 11a and 11b**). Through assessment of expression of circRIG-I in colon cancer tissues and their matched adjacent normal tissues, we found that circRIG-I was upregulated in colon cancers (**Response Fig. 11c**). Further analysis of the clinic-pathological features revealed that ulcerative colon cancers expressed higher level of circRIG-I than polypoid colon cancers did (**Response Fig. 11d**). Moreover, upregulation of circRIG-I was highly related with cancer progression and patients' age, while no significant correlation was observed concerning patients' gender, tumor subtype and location (**Response Fig. 11e-i**). These findings thus indicated that dysregulation of circRIG-I may contribute to colon cancer progression.

a

b

c

d

e

f

g

h

i

Response Figure 11. Clinical analysis of circRIG-I expression in colon cancers.

(a) Human circRIG-I can be interrogated by the CIRCpedia V2 database (<http://yang-laboratory.com/circpedia/search>).

(b) Sequence of human circRIG-I junction site was shown in the colon cancer sample.

(c) RT-qPCR analysis of the relative level of circRIG-I in colon cancers and their matched adjacent normal tissues (n = 100, **** $P < 0.0001$, two-tailed paired Student's t-test).

(d) RT-qPCR analysis of the relative level of circRIG-I between ulcerative colon cancer (n = 44) and polypoid (n = 30) colon cancers (* $P < 0.05$, two-tailed unpaired Student's t-test).

(e) RT-qPCR analysis of the relative level of circRIG-I in colon cancers with different stage (* $P < 0.05$, two-tailed unpaired Student's t-test).

(f) RT-qPCR analysis of the relative level of circRIG-I in colon cancers with different

age (T1/T2, n = 14; T3, n = 76; T4, n = 10; ** $P < 0.01$, two-tailed unpaired Student's t-test).

(g) RT-qPCR analysis of the relative level of circRIG-I between colon adenocarcinoma (n = 87) and mucinous colon adenocarcinoma (n = 13) (ns, not significant ($P > 0.05$), two-tailed unpaired Student's t-test).

(h) RT-qPCR analysis of the relative level of circRIG-I in colon cancers with different gender (male, n = 53; female, n = 47; ns, not significant ($P > 0.05$), two-tailed unpaired Student's t-test).

(i) RT-qPCR analysis of the relative level of circRIG-I in colon cancers with different locations (colon, n = 50; rectum, n = 47; ileocecus, n = 3; ns, not significant ($P > 0.05$), two-tailed unpaired Student's t-test).

The **Response Figure 11b-i** were also shown in **Fig. 5a-h** in revised version.

2. There is no data presented to show if the circular RNA identified in the mouse model is conserved to human. Accordingly, it was also not shown if colon cancer patients with DDX58/RIG-I frameshift mutations have elevated levels of circRIG-I. These aspects are of key importance to link the findings from the mouse model to the discovery of the germline DDX58/RIG-I frameshift mutations in patients with colon cancer.

[Response] We thank the reviewer for this question. The murine circRIG-I is conserved to human (<http://yang-laboratory.com/circpedia/search>), which was further confirmed by our experiments (**Response Fig. 12a and 12b**). Through assessment of expression of circRIG-I in colon cancer tissues and their matched adjacent normal tissues, we found that circRIG-I was upregulated in colon cancers (**Response Fig. 12c**). Of note, the expression of circRIG-I was significantly increased in cancers bearing frameshift variants as relative to their matched adjacent normal tissues (**Response Fig. 12d**), which further support the notion that frameshift mutation of *RIG-I* enhances the upregulation of circRIG-I.

Response Figure 12. CircRIG-I is upregulated in colon cancers.

(a) Human circRIG-I can be interrogated by the CIRCpedia V2 database (<http://yang-laboratory.com/circpedia/search>).

(b) Sequence of human circRIG-I junction site was shown in the colon cancer sample.

(c) RT-qPCR analysis of the relative level of circRIG-I in colon cancers and their matched adjacent normal tissues (n = 100; **** $P < 0.0001$, two-tailed paired Student's t-test).

(d) RT-qPCR analysis of the relative level of circRIG-I in colon cancers and their matched adjacent normal tissues from patients with frameshift mutation of *RIG-I* (n = 3; * $P < 0.05$, two-tailed paired Student's t-test).

The **Response Figure 12b and 12c** were also shown in **Fig. 5a,b** in the revised version.

3. For the overexpression experiments of circRIG-I, the authors should do a Northern blotting +/- RNase R treatment to provide information about whether the vector produces concatemers and linear co-products. Also, the overexpression seems to be very efficient. How do the levels compare to endogenous levels? The authors should comment on whether the levels can be considered physiologically relevant.

[Response] As suggested, we used probes that hybridize with the splicing junction to distinguish circRIG-I and probes that hybridize with exon 5 to distinguish circRIG-I and its host gene, RIG-I, by northern blotting (**Response Fig. 13a**), which further confirmed that existence of circRIG-I. In addition, we compared the expression levels of endogenous and exogenous circRIG-I. As shown in **Response Figure 13b**, the level of exogenous overexpression of circRIG-I is comparable to the endogenous level of circRIG-I in *Rig-1^{fs/fs}* MEFs.

Response Figure 13. Inducible circRIG-I is observed in *Rig-1^{fs/fs}* MEFs.

(a) Northern blotting analysis of circRIG-I and RIG-I mRNA levels by hybridization with exon 5 (upper) and exon 5-exon 12 junction (lower) probes with and without RNase R treatment. ACTB mRNA with or without RNase R treatment was detected as a control.

(b) Conventional PCR and RT-qPCR analysis of the expression levels of endogenous and exogenous circRIG-I in indicated cells.

The **Response Figure 13a** was also shown in **Fig. 4k** in revised version.

4. Did the authors BLAST the shRNA sequences in a search for potential off-target effects?

[Response] Yes, we have queried the shRNA sequences using NCBI BLAST servers for identifying potential off-target sites. Through screening, we finally used four short hairpin RNAs (shRNAs) that target the back-splice junction of circRIG-I to selectively knockdown circRIG-I without affecting its linear counterpart *Rig-i*.

Minor revisions

1. Regarding the Knock down experiments of circRIG-I, some of the used shRNAs are quite skewed on the back-splicing junction (indicated by |): (1#: CTCGGAGGTTTGAAG|GTTTCA; 2#: GAGGTTTGAAG|GTTTCAAAG; 3#: GTTTGAAG|GTTTCAAAGGGC; 4#: GAAG|GTTTCAAAGGGCTGAA). It was probably expected based on this that #1 and #4 knocked down the linear counterpart. The authors might want to comment on this.

[Response] Yes, the reason that we did not use 1[#] and 4[#] was that they had inhibitory effects on the linear *Rig-i* mRNA. As suggested, we have mentioned this reason in the revised version.

2. The authors might want to comment on that frameshift germline mutations in DDX58/RIG-I have been observed before according to the COSMIC database (https://cancer.sanger.ac.uk/cosmic/gene/analysis?all_data=&coords=AA%3AAA&dr=&end=926&gd=&id=287669&ln=DDX58&seqLen=926&sn=large_intestine&start=1#t). These data seem to be in accordance with the present findings and this could, for instance, be mentioned in the discussion.

[Response] We thank the reviewer for this suggestion. We have noticed the mutation of RIG-I (Mutation ID: COSM6995308) in the COSMIC database. As suggested, we have mentioned this mutation in the discussion.

3. The authors show that ATRA betters the cancer-related inflammation by suppression of circRIG-I expression, through promotion of DHX9. Since DHX9 is involved in the splicing and alternative splicing of many genes, could it be speculated that it has other effects and/or side-effects. They authors might want to comment on this in the discussion section.

[Response] As suggested, we have discussed other potential effects and/or side-effects of ATRA in the revised version.

4. Throughout there are sentences that have grammatical errors, please read the entire manuscript to carefully to address this.

[Response] We appreciate the reviewer's careful reading of our manuscript. We have corrected the grammatical errors in the revised version.

Reviewer #4 (Remarks to the Author): with expertise in colitis, colorectal cancer

In the manuscript "Mutant RIG-I enhances cancer related inflammation through activation of circRIG-I signaling" Song et al. report on numerous findings based on a RIG-I variant.

The manuscript contains numerous experiments of different fields of innate signaling (cancer, inflammation, viral response). For me it is not clear why the authors want to connect this RIG-I variant to cancer? There is very limited data on cancer-related aspects in this manuscript. Detecting 3 out of 350 sporadic CRC patients with a variant is certainly too weak to make this claim.

[Response] Because the non-specific inflammatory lesions contribute to the malignant transformation of normal cells. Previous study reveals that RIG-I expression is downregulated in human hepatocellular carcinoma (HCC). Through activation of type-I interferon signaling, RIG-I acts as a tumor suppressor that is closely related with tumor

development and the formation of tumor microenvironment. To assess the status of *RIG-I* in colon cancers, we performed sequencing analysis of the *RIG-I* exons in 227 blood samples from patients with colon cancer and identified three patients with heterozygous frameshift variants at residue 395 in exon 3 (two samples with T³⁹⁵>- and one sample with TTAA³⁹⁵⁻³⁹⁸>-). This incidence of *RIG-I* mutation was significantly higher than the population frequency (1.32% vs. 0.0004%) interrogated from GnomAD database, which is accordance with our data that no *RIG-I* mutation was detected in 350 healthy persons. During the revision phase, we identified another two cases bearing these frameshift variants at residue 395 in exon 3 from 198 blood samples from patients with colon cancer, no frameshift variants of *RIG-I* were detected in healthy control subjects (**Response Fig.14a and 14b**).

Furthermore, we also assessed the expression of circRIG-I in colon cancer tissues as well as adjacent normal tissues (**Response Fig.14c**). As shown in **Response Figure 14d and 14e**, circRIG-I was upregulated in colon cancer tissues. Of note, the expression of circRIG-I was significantly increased in cancers bearing frameshift variants as relative to their matched adjacent normal tissues, suggesting the involvement of circRIG-I in tumorigenesis. Due to the essential role of *RIG-I* in host innate immunity, we thus used viral infection model to assess the effects of circRIG-I on inflammatory response.

Response Figure 14. Frameshift mutation of *RIG-I* increases circRIG-I expression in colon cancer.

(a-b) Sequence of frameshift mutation of *RIG-I* in patients with colon cancer.

(c) Sequencing analysis of human circRIG-I junction site was shown in the colon cancer sample.

(d) RT-qPCR analysis of the relative level of circRIG-I in colon cancers and their matched adjacent normal tissues (n = 100; *****P* < 0.0001, two-tailed paired Student's t-test).

(e) RT-qPCR analysis of the relative level of circRIG-I in colon cancers and their matched adjacent normal tissues from patients with frameshift mutation of *RIG-I* (n =

3; * $P < 0.05$, two-tailed paired Student's t-test).

The **Response Figure 14a and 14b** as well as **Response Figure 14c and 14d** were also shown in **Extended Data Fig. 1** and **Fig. 5a,b** in the revised version, respectively.

In addition using a very weak animal model for CAC is not sufficient per se and in particular that the effect of the RIG-1 variant could only be demonstrated on the inflammatory response. Did the authors find the variant in ulcerative colitis patients.

[Response] As suggested, we performed sequencing analysis of the *RIG-I* exons in 77 blood samples from patients with ulcerative colitis and identified one patient with heterozygous frameshift variant at residue 395 in exon 3 (TTAA³⁹⁵⁻³⁹⁸>-) (**Response Fig. 15a**). In addition, we analyzed the expression of circRIG-I in the inflammatory foci (P) and matched adjacent normal tissues (N) from patients with ulcerative colitis, we found that circRIG-I was upregulated in the inflammatory foci (**Response Figure 15b**). Our data thus indicate that circRIG-I is involved in tumor-related inflammation.

Response Figure 15. CircRIG-I is upregulated in the inflammatory foci from patients with ulcerative colitis.

(a) Sequence of frameshift mutation of *RIG-I* in patients with ulcerative colitis.

(b) RT-qPCR analysis of the relative level of circRIG-I in the inflammatory foci (P) and matched adjacent normal tissues (N) from patients with ulcerative colitis (n = 8; ** $P < 0.01$, two-tailed paired Student's t-test).

The **Response Figure 15b** was also shown in **Fig. 5i** in the revised version.

The authors do not investigate how the variant is distributed among the CMS types of CRC.

[Response] As suggested, we have presented the clinical information in the **Extended Data Table 3**. Though analysis of the patients carrying the frameshift mutations, we found that all tumors were ulcerative adenocarcinoma of colon (pT4aN1a, pT3N0 and pT3aN1a). In addition, we also identified human circRIG-I that is conserved to mice (**Response Fig. 16a and 16b**). Through assessment of expression of circRIG-I in colon cancer tissues and their matched adjacent normal tissues, we found that circRIG-I was upregulated in colon cancers (**Response Fig. 16c**). Compared with polypoid colon cancers, ulcerative colon cancers expressed higher level of circRIG-I (**Response Fig. 16d**). Moreover, upregulation of circRIG-I was highly correlated with cancer progression and age (**Response Fig. 16e and 16f**). We also studied other information and found that gender, subtype and location were not related with circRIG-I induction (**Response Fig. 16g-i**).

a

circID	Species	Gene	Isoform	Location	Strand	FPM	ExonStart-ExonEnd	Seq Type	Cell Line	Conservation	Mapsplice	Enrichment
HSA_CIRCpedia_190325	Human (hg38)	DDX58	NM_014314	Chr9:32480218-32491420	-	0.008	5-12	Ribo-	Liver_N17	MMU_CIRCpedia_216979	No	N.A.

b

c

d

e

f

g

h

i

Response Figure 16. Clinical analysis of circRIG-I expression in colon cancers.

(a) Human circRIG-I can be searched by the CIRCpedia V2 database (<http://yang-laboratory.com/circpedia/search>).

(b) Sequence of human circRIG-I junction site was shown in the colon cancer sample.

(c) RT-qPCR analysis of the relative level of circRIG-I in colon cancers and their matched adjacent normal tissues (n = 100, **** $P < 0.0001$, two-tailed paired Student's t-test).

(d) RT-qPCR analysis of the relative level of circRIG-I between ulcerative colon cancer (n = 44) and polypoid (n = 30) colon cancers (* $P < 0.05$, two-tailed unpaired Student's t-test).

(e) RT-qPCR analysis of the relative level of circRIG-I in colon cancers with different stage (* $P < 0.05$, two-tailed unpaired Student's t-test).

(f) RT-qPCR analysis of the relative level of circRIG-I in colon cancers with different

age (T1/T2, n = 14; T3, n = 76; T4, n = 10; ** $P < 0.01$, two-tailed unpaired Student's t-test).

(g) RT-qPCR analysis of the relative level of circRIG-I between colon adenocarcinoma (n = 87) and mucinous colon adenocarcinoma (n = 13) (ns, not significant ($P > 0.05$), two-tailed unpaired Student's t-test).

(h) RT-qPCR analysis of the relative level of circRIG-I in colon cancers with different gender (male, n = 53; female, n = 47; ns, not significant ($P > 0.05$), two-tailed unpaired Student's t-test).

(i) RT-qPCR analysis of the relative level of circRIG-I in colon cancers with different locations (colon, n = 50; rectum, n = 47; ileocecus, n = 3; ns, not significant ($P > 0.05$), two-tailed unpaired Student's t-test).

The **Response Figure 16b-i** were also shown in **Fig. 5a-h** in revised version.

Whereas the experiments are well done the manuscript lacks focus completely. I cannot see that the tumor microenvironment of colon cancer is altered based on the variant. I would suggest to restructure the complete manuscript and focus on a cancer-independent inflammatory response of substantiate the cancer information in the manuscript.

[Response] We thank the reviewer for the comments. Because the non-specific inflammatory lesions contribute to the malignant transformation of normal cells. Considering the key role of RIG-I in host innate immunity, we thus assessed the status of RIG-I in patients with colon cancer and identified three patients with heterozygous frameshift variants at residue 395 in exon 3 (two samples with T³⁹⁵>- and one sample with TTAA³⁹⁵⁻³⁹⁸>-). This incidence of *RIG-I* mutation was significantly higher than the population frequency (1.32% vs. 0.0004%) interrogated from GnomAD database, which is accordance with our data that no *RIG-I* mutation was detected in 350 healthy persons. During the revision phase, we identified another two cases bearing these frameshift variants at residue 395 in exon 3 from 198 blood samples from patients with colon cancer, no frameshift variants of *RIG-I* were detected in healthy control subjects. Given the increased risk of development of colon cancer in patients with ulcerative colitis, we also performed the sequencing assay to assess the status of RIG-I in patients with colitis. We identified that this kind frameshift mutation of *RIG-I* (TTAA³⁹⁵⁻³⁹⁸>-) was occurred in one patient with ulcerative colitis (1/77).

Mechanistically, the frameshift mutation of *RIG-I* not only interrupted the translation of RIG-I but also induced circRIG-I expression. Accordingly, the expression of circRIG-I was increased in colon cancer tissues as compared with adjacent normal tissues, supporting the notion that the frameshift mutation of *RIG-I* is implicated in colon cancer development. Further clinical analysis revealed that high expression of circRIG-I was preferentially detected in progressive ulcerative colon cancer. In light of the stimulatory effects of circRIG-I on inflammation, we thus hypothesize that the frameshift mutation of *RIG-I* exacerbates colon cancer development through activating circRIG-I signaling.

Reviewers' comments:

Reviewer #1 (Remarks to the Author):

In the revised version of their manuscript "Mutant RIG-I enhances cancer-related inflammation through activation of circRIG-I signaling" Song et al. provide some additional data but to me the fundamental doubts on the correctness of the mechanistic explanations persist.

1. I do not believe that the phenotype of the RIG-I FS/FS mutation is explained by an increased amount of the circ. RIG-I RNA:

a. As already stated the circRIG-I RNA is not structurally changed in any way by the mutation – the argument is purely based on a quantitative change.

The DDX58/RIG-I gene is a classical interferon-induced gene and when the (unmutated) gene is induced - in parallel to the induction of the linear RIG-I mRNA - the amount of circularRIG-I RNA is increased. As can be seen e.g. in response figure 14d and 14c the amount of circRNA is up to 10 times higher in unmutated tumors than in mutated tumors (60 fold versus 6) and e.g. in figure 4m it can be seen that for circRIG the induction by the VSV-infection in the WT makes up the main quantitative difference from below 4 to about 80 fold. Compared to the effect of the virus infection the RIG-I FS/FS mutation then adds only a "small" 2 fold additional shift from 80 to 160. (as a side note: the legend to figure 4m/n does not explain what is used as the basis for the fold-induction – please complement the legend to make this understandable)

So every interferon-signal triggers circRIG-I – but does not show the phenotype of the FS/FS mouse.

b. Within the induced circular RNAs the circRIG-I is a random pick – the argument that it plays a special role because it is localized on chromosome 4 to me does not make sense

c. The assay (figure 4h) arguing for a different binding of wt and RIG-I FS/FS Pre-mRNA to DHX9 is not described in the figure legend or the method section in a way that one can really understand how the assay was done (what RNAs were used? how were they prepared? By in vitro transcription? how was the pulldown done...). But if I imagine correctly how this assay is done this is a very error prone assay to measure quantitative differences. Therefore, I think the evidence for difference in DHX9-binding is also weak.

d. There is a discrepancy between where the DHX9 binding sites are described in figure 4g – (there does not seem to be anything special about exome 3) and where the premRNA structural differences induced by the FS/FS mutation are predicted (= around exome 3).

2. Song et al. provide in their revised version of the manuscript new data concerning the question if the truncated RIG-I generated by the FS/FS mutation is expressed and explains the phenotype or not. However I do not share their interpretation: They show that if over-expressed in form of a plasmid the truncated protein is expressed and even though in their in vitro assay not stimulatory by overexpression alone at least seems to have an influence on MAVS activation by intact RIG-I.

In figure 4 k they show by northernblot the existence of RIG-I mRNA and circularRNA but only in WT MEFs and not in the FS/FS MEFs – which would have been the experiment suggested. But by PCR assay at least they do see the expression of the RIG-I mRNA also in FS/FS MEFs. To me this makes it very likely that the shortened protein actually is expressed. The non-detection by westernblot or mass spectrometry I guess is a quantitative problem and does not exclude the expression. In the response figure 1 b, the given peptides and position do not match but besides this error only the first 2 peptides could theoretically be present in the shortend RIG-I version and these would both be localized at the often underrepresented ends of the protein. I therefore do not take the non-detection of these peptides by mass-spectrometry as a proof that the protein is not present.

The sh-RNA experiments do fit to the prediction of a RIG-I gain of function protein. Much of the effect is lost by MAVS depletion but not by MDA-5 depletion. That the effect is gone in DDX3X and IRF3 can be interpreted that these two proteins are relevant and required for the downstream signalling. But I do not think that this experiment proves that DDX3X is an interferon-inducing sensor of circRIG-I. The data of DDX3X as a genuine sensor are so far scarce and I think more people believe DDX3X to be rather a scaffolding protein that enhances a signal triggered via RIG-I or MDA-5 via MAVS then to be a real independent sensor that triggers MAVS-dependent signaling independent of the RLRs.

The sh-Experiments that try to knock-down circRIG-I without targeting the RIG-I mRNA – to me still

lack specificity (I do not believe that the shRNAs 2 and 3 here have the specificity that 1 and 4 did not have). Using pppRNA as specific RIG-I ligand as a control might help to show that the function of the RIG-I protein is really unchanged.

3. I would ask to make clear in figures that use human cell lines what construct are used - are they of murine or human origin. e.g. RIG-I 1-284, the overexpressed circRIG-I ect.

Reviewer #2 (Remarks to the Author):

The authors have performed additional experiments to support their conclusion. I have no other concern.

Reviewer #3 (Remarks to the Author):

The authors have responded well to all of my previous comments and the paper has been significantly improved.

The response to reviewers is listed as follows:

Reviewer #1 (Remarks to the Author):

In the revised version of their manuscript “Mutant RIG-I enhances cancer-related inflammation through activation of circRIG-I signaling” Song et al. provide some additional data but to me the fundamental doubts on the correctness of the mechanistic explanations persist.

1. I do not believe that the phenotype of the RIG-I FS/FS mutation is explained by an increased amount of the circ. RIG-I RNA:

a. As already stated the circRIG-I RNA is not structurally changed in any way by the mutation – the argument is purely based on a quantitative change.

The DDX58/RIG-I gene is a classical interferon-induced gene and when the (unmutated) gene is induced - in parallel to the induction of the linear RIG-I mRNA - the amount of circular RIG-I RNA is increased. As can be seen e.g. in response figure 14d and 14c the amount of circRNA is up to 10 times higher in unmutated tumors than in mutated tumors (60 fold versus 6) and e.g. in figure 4m it can be seen that for circRIG the induction by the VSV-infection in the WT makes up the main quantitative difference from below 4 to about 80 fold. Compared to the effect of the virus infection the RIG-I FS/FS mutation then adds only a “small” 2 fold additional shift from 80 to 160. (as a side note: the legend to figure 4m/n does not explain what is used as the basis for the fold-induction – please complement the legend to make this understandable)

So every interferon-signal triggers circRIG-I – but does not show the phenotype of the FS/FS mouse.

[Response] Indeed, our legend for **Figure 4m** has mention that we used the WT and *Rig-i*^{fs/fs} MEFs to analyze the expression of circRIG-I. Additionally, we have labelled the tissues we analyzed in **Figure 4n (Response Fig. 1a)**.

Although circRIG-I expression is upregulated upon exposure to IFN treatment, frameshift mutation of RIG-I enhances circRIG-I expression with or without viral infection (**Response Fig. 1b**). Moreover, this frameshift mutation of RIG-I selectively impaired the stem loop structure and resulted in the dissociation of *Rig-i* mRNA with DHX9, consequently leading to upregulation of circRIG-I (**Response Fig. 1c**). In accordance with the *in vitro* data, we found that tumors bearing frameshift mutation of RIG-I expressed high level of circRIG-I, which is highly correlated with poor outcome of cancer patients (**Response Fig. 1d-g**). Our data thus suggest that frameshift mutation of *RIG-I* promotes colon cancer development through induction of circRIG-I.

Response Figure 1. Mutant *RIG-I* promotes colon cancer development through activation of circRIG-I signaling.

(a) RT-qPCR analysis of circRIG-I expression in indicated tissues from wild-type (WT) and *Rig-1^{fs/fs}* mice infected with VSV (n = 5 biological replicates, mean ± s.e.m., ***P* < 0.01, two-tailed paired Student's t-test).

(b) RT-qPCR analysis of circRIG-I expression in wild-type (WT) and *Rig-1^{fs/fs}* MEFs with VSV or LCMV C113 infection (n = 3 biological replicates, mean ± s.e.m., **P* < 0.05, ***P* < 0.01, two-tailed paired Student's t-test).

(c) RNA secondary structures of wild-type (WT) and frameshift mutant *RIG-I* pre-mRNA were predicted by RNAfold web server (<http://rna.tbi.univie.ac.at/cgi-bin/RNAWebSuite/RNAfold.cgi>). The sequences of *Rig-i* exon 3 and intron 3 were analyzed. The frameshift mutation was highlighted in red color and the pre-mRNA structural change was highlighted in blue color.

(d) RT-qPCR analysis of the relative level of circRIG-I in colon cancers and their matched adjacent normal tissues from patients with frameshift mutation of *RIG-I* (n = 3; **P* < 0.05, two-tailed paired Student's t-test).

(e) RT-qPCR analysis of the relative level of circRIG-I between ulcerative colon cancer (n = 44) and polypoid (n = 30) colon cancers (**P* < 0.05, two-tailed unpaired Student's t-test).

(f) RT-qPCR analysis of the relative level of circRIG-I in colon cancers with different stage (T1/T2, n = 14; T3, n = 76; T4, n = 10; **P* < 0.05, two-tailed unpaired Student's

t-test).

(g) RT-qPCR analysis of the relative level of circRIG-I in colon cancers with different age (< 65, n = 52; ≥ 65, n = 48; ** $P < 0.01$, two-tailed unpaired Student's t-test).

b. Within the induced circular RNAs the circRIG-I is a random pick – the argument that it plays a special role because it is localized on chromosome 4 to me does not make sense

[**Response**] We firstly used the RNAfold web server to predict RNA secondary structure and found that frameshift mutation significantly impaired the stem loop structure relative to that in wildtype RNA (**Response Fig. 2a**). Interestingly, this region containing the stem loop, has been reported to be essential for DHX9-binding (**Response Fig. 2b**). Ensued RNA-pulldown assay further confirmed the relationship between pre-mRNA of *Rig-i* and DHX9 (**Response Fig. 2c**). In accordance with our hypothesis, the frameshift mutation of *Rig-I* remarkably impaired its association with DHX9 (**Response Fig. 2c**). In light of the essential role of DHX9 in modulation of circRNA biogenesis, we thus hypothesize that the frameshift mutation of *Rig-I* can trigger circRNA generation from the RIG-I transcript. In order to reduce the background noise and increase the reliability and accuracy of circRNA identification, we treated ribosomal RNA-depleted total RNAs from virus-infected WT and *Rig-i*^{fs/fs} MEFs with RNase R to degrade linear RNAs and enrich circRNAs, eventually subjected to high-throughput RNA sequencing. Finally, we identified the mmu-DDX58_0004 (circAtlas ID) (<http://circatlas.biols.ac.cn/>), which was selectively upregulated in *Rig-i*^{fs/fs} MEFs as compared with WT control. Considering the dissociation of *Rig-i* mRNA with DHX9, we thus assume that frameshift mutation triggers circRIG-I upregulation.

Response Figure 2. *CircRIG-I* is selectively induced by mutant *RIG-I* during viral infection.

(a) RNA secondary structures of wild-type (WT) and frameshift mutant *RIG-I* pre-mRNA were predicted by RNAfold web server (<http://rna.tbi.univie.ac.at/cgi-bin/RNAWebSuite/RNAfold.cgi>). The sequences of *Rig-i* exon 3 and intron 3 were analyzed. The frameshift mutation was highlighted in red color and the pre-mRNA structural change was highlighted in blue color.

(b) Integrative Genomics Viewer (IGV) analysis of CLIP-seq coverage, DHX9 peaks at the Alu elements (SINE) of *Ddx58* were highlighted. Data range represents the coverage of uniquely mapped alignments for each profile.

(c) Interaction between DHX9 and wild-type (WT) or frameshift mutant *RIG-I* pre-mRNA. HEK293T cells were transfected with FLAG-DHX9 expressing vector. The biotin-labeled WT or frameshift mutation of murine *Rig-i* RNA including exon 3 and intron 3 was obtained by T7 RNA polymerase kit and Biotin RNA Labeling Mix. WT *Rig-i* RNA antisense sequence was used as control. Labeled RNA was incubated with cell lysates from HEK293T cells and RNA binding proteins were isolated by streptavidin agarose beads. The binding components were eluted by biotin, followed by SDS-Page.

c. The assay (figure 4h) arguing for a different binding of wt and RIG-I FS/FS Pre-mRNA to DHX9 is not described in the figure legend or the method section in a way that one can really understand how the assay was done (what RNAs were used? how were they prepared? By in vitro transcription? how was the pulldown done...). But if I imagine correctly how this assay is done this is a very error prone assay to measure quantitative differences. Therefore, I think the evidence for difference in DHX9-binding is also weak.

[**Response**] As suggested, we have added the method related with *in vitro* RNA pulldown assay. Briefly, the biotin-labeled WT or frameshift mutation of murine *Rig-i* RNA including exon 3 and intron 3 was obtained by T7 RNA polymerase kit and Biotin RNA Labeling Mix. WT *Rig-i* RNA antisense sequence was used as control. Labeled RNA was incubated with cell lysates from HEK293T cells overexpressing DHX9-FLAG and RNA binding proteins were isolated by streptavidin agarose beads. The binding components were eluted by biotin, followed by SDS-Page. We found that the frameshift mutation of *Rig-i* remarkably impaired its association with DHX9.

d. There is a discrepancy between where the DHX9 binding sites are described in figure 4g – (there does not seem to be anything special about exome 3) and where the premRNA structural differences induced by the FS/FS mutation are predicted (= around exome 3).

[**Response**] It has been reported that DHX9 binds specifically to the short interspersed nuclear element (SINE) that are transcribed as parts of genes. Accordingly, the published cross-linking immunoprecipitation sequencing (CLIP-seq) data of DHX9 showed that the DHX9-binding sites mainly enriched in SINE-enriched region from Exon 3 to Exon 5 and region from Exon 12 to Exon 15, respectively. Of note, the region from Exon 3 to Intron 3 contains the DHX9-binding peaks and SINE (**Response Fig. 3a**). Utilizing the RNAfold web server to predict RNA secondary structure, we found that frameshift mutation occurred in Exon 3 significantly impaired the stem loop structure that is localized in Intron 3 (**Response Fig. 3b**). In light of the dissociation of *Rig-i* mRNA with DHX9 (**Response Fig. 3c**), our data thus indicate that the secondary structure of mRNA (the stem loop) is critical for DHX9-binding in addition to its primary structure (SINE sequence).

Response Figure 3. Frameshift mutation of RIG-I changes the secondary structure of RIG-I pre-mRNA and impairs its association with DHX9.

(a) RNA secondary structures of wild-type (WT) and frameshift mutant *RIG-I* pre-mRNA were predicted by RNAfold web server (<http://rna.tbi.univie.ac.at/cgi-bin/RNAWebSuite/RNAfold.cgi>). The sequences of *Rig-i* exon 3 and intron 3 were analyzed. The frameshift mutation was highlighted in red color and the pre-mRNA structural change was highlighted in blue color.

(b) Integrative Genomics Viewer (IGV) analysis of CLIP-seq coverage, DHX9 peaks at the Alu elements (SINE) of *Ddx58* were highlighted. Data range represents the coverage of uniquely mapped alignments for each profile.

(c) Interaction between DHX9 and wild-type (WT) or frameshift mutant *RIG-I* pre-mRNA. HEK293T cells were transfected with FLAG-DHX9 expressing vector. The biotin-labeled WT or frameshift mutation of murine *Rig-i* RNA including exon 3 and intron 3 was obtained by T7 RNA polymerase kit and Biotin RNA Labeling Mix. WT *Rig-i* RNA antisense sequence was used as control. Labeled RNA was incubated with cell lysates from HEK293T cells and RNA binding proteins were isolated by streptavidin agarose beads. The binding components were eluted by biotin, followed by SDS-Page.

2. Song et al. provide in their revised version of the manuscript new data concerning the question if the truncated RIG-I generated by the FS/FS mutation is expressed and explains the phenotype or not. However I do not share their interpretation: They show that if over-expressed in form of a plasmid the truncated protein is expressed and even though in their in vitro assay not stimulatory by overexpression alone at least seems to have an influence on MAVS activation by intact RIG-I.

[Response] To answer the reviewer’s question that “A potentially much easier explanation for the phenotype would be if the frameshift mutation leads to a translated protein with a gain of function phenotype”, we thereby cloned the frameshift mutant *RIG-I* into pEGFP-C1 vector and determine whether the predicted truncated RIG-I elicited effects on host innate immunity. Unlike the active truncation of RIG-I

(RIG-I₁₋₂₈₄), the predicted truncated RIG-I (RIG-I₁₋₁₄₀) hardly affected host innate immunity under quiescent condition (**Response Fig. 4a**). Moreover, when the predicted truncated RIG-I was co-transfected with human-derived RIG-I or active truncated RIG-I (RIG-I₁₋₂₈₄) into HEK293T cells. The predicted truncated RIG-I (RIG-I₁₋₁₄₀) elicited inhibitory effects on host antiviral immunity (**Response Fig. 4b,c**), which cannot explain the phenotype of *Rig-1^{fs/fs}* mice. Our data thus indicate that other mechanisms contribute to the antiviral effects induced by the frameshift mutation of *RIG-I*.

Response Figure 4. The predicted RIG-I truncation exerts inhibitory effects on host innate immune response.

(a) RT-qPCR analysis of the RNA level of SeV in HEK293T cells transfected with indicated vectors following SeV infection (n = 3 biological replicates, mean \pm s.e.m.). (b-c) HEK293T cells were transfected with indicated plasmids and then infected with VSV-GFP for 12 hours. Images of GFP⁺ cells were shown in (b). Flow cytometric analysis of GFP⁺ cells were shown in (c) (n = 3 biological replicates, mean \pm s.e.m., ****P* = 0.0001, *****P* < 0.0001, unpaired Student's t-test).

In figure 4 k they show by northern blot the existence of RIG-I mRNA and circularRNA but only in WT MEFs and not in the FS/FS MEFs – which would have been the experiment suggested. But by PCR assay at least they do see the expression of the RIG-I mRNA also in FS/FS MEFs. To me this makes it very likely that the shortened protein actually is expressed. The non-detection by western blot or mass spectrometry I guess is a quantitative problem and does not exclude the expression.

[Response] Recent study reveals that genetic mutations usually affect the composition of the transcribed mRNA and its encoded protein, leading to instability of the mRNA and/or the protein (PMID: 33971252). Our present study found that the frameshift mutation of RIG-I reduced the mRNA level of *RIG-I* and impaired its translation into

protein. Furthermore, we cloned the frameshift mutant *RIG-I* into pEGFP-C1 vector and determine whether the predicted truncated RIG-I elicited effects on host innate immunity. Unlike the active truncation of RIG-I (RIG-I₁₋₂₈₄), the predicted truncated RIG-I (RIG-I₁₋₁₄₀) hardly affected host innate immunity under quiescent condition (**Response Fig. 5a**). Moreover, when the predicted truncated RIG-I (RIG-I₁₋₁₄₀) was co-transfected with RIG-I or active truncated RIG-I (RIG-I₁₋₂₈₄) into HEK293T cells. The predicted truncated RIG-I (RIG-I₁₋₂₈₄) elicited inhibitory effects on host antiviral immunity (**Response Fig. 5b,c**), which cannot explain the phenotype of *Rig-1^{fs/fs}* mice. Our data thus indicate that other mechanisms contribute to the antiviral effects induced by the frameshift mutation of *RIG-I*.

Additionally, considering the facts that full length of RIG-I rather than truncated RIG-I can be detected in cells by mass spectrometry and antibody, we thus ruled out the possibility of a quantitative problem.

Response Figure 5. The predicted RIG-I truncation exerts inhibitory effects on host innate immune response.

(a) RT-qPCR analysis of the RNA level of SeV in HEK293T cells transfected with indicated vectors following SeV infection (n = 3 biological replicates, mean \pm s.e.m.). (b-c) HEK293T cells were transfected with indicated plasmids and then infected with VSV-GFP for 12 hours. Images of GFP⁺ cells were shown in (b). Flow cytometric analysis of GFP⁺ cells were shown in (c) (n = 3 biological replicates, mean \pm s.e.m., ****P* = 0.0001, *****P* < 0.0001, unpaired Student's t-test).

In the response figure 1 b, the given peptides and position do not match but besides this error only the first 2 peptides could theoretically be present in the shortend RIG-I version and these would both be localized at the often underrepresented ends of the protein. I therefore do not take the non-detection of these peptides by mass-spectrometry as a proof that the protein is not present.

[Response] We regret this error and we have replaced it with right figure as shown in **Response Figure 6a**. Recent study reveals that genetic mutations usually affect the composition of the transcribed mRNA and its encoded protein, leading to instability of the mRNA and/or the protein (PMID: 33971252). Our present study found that the frameshift mutation of RIG-I reduced the mRNA level of *RIG-I* and impaired its translation into protein. Furthermore, we cloned the frameshift mutant *RIG-I* into pEGFP-C1 vector and determine whether the predicted truncated RIG-I (RIG-I₁₋₁₄₀) elicited effects on host innate immunity. Unlike the active truncation of RIG-I (RIG-I₁₋₂₈₄), the predicted truncated RIG-I (RIG-I₁₋₁₄₀) hardly affected host innate immunity under quiescent condition (**Response Fig. 6b**). Moreover, when the predicted truncated RIG-I (RIG-I₁₋₁₄₀) was co-transfected with RIG-I or active truncated RIG-I (RIG-I₁₋₂₈₄) into HEK293T cells. The predicted truncated RIG-I (RIG-I₁₋₁₄₀) elicited inhibitory effects on host antiviral immunity (**Response Fig. 6c,d**), which cannot explain the phenotype of *Rig-i^{fs/fs}* mice. Our data thus indicate that other mechanisms contribute to the antiviral effects induced by the frameshift mutation of *RIG-I*.

Response Figure 6. The predicted RIG-I truncation exerts inhibitory effects on host innate immune response.

(a) Mass spectrometry analysis of RIG-I expression in wild-type and *Rig-i^{fs/fs}* MEFs. Protein peptides detected in wild-type MEFs were shown.

(b) RT-qPCR analysis of the RNA level of SeV in HEK293T cells transfected with indicated vectors following SeV infection (n = 3 biological replicates, mean \pm s.e.m.).

(c-d) HEK293T cells were transfected with indicated plasmids and then infected with VSV-GFP for 12 hours. Images of GFP⁺ cells were shown in (c). Flow cytometric analysis of GFP⁺ cells were shown in (d) (n = 3 biological replicates, mean \pm s.e.m., ***P = 0.0001, ****P < 0.0001, unpaired Student's t-test).

The sh-RNA experiments do fit to the prediction of a RIG-I gain of function protein. Much of the effect is lost by MAVS depletion but not by MDA-5 depletion. That the effect is gone in DDX3X and IRF3 can be interpreted that these two proteins are relevant and required for the downstream signalling. But I do not think that this experiment proves that DDX3X is an interferon-inducing sensor of circRIG-I. The data of DDX3X as a genuine sensor are so far scarce and I think more people believe DDX3X to be rather a scaffolding protein that enhances a signal triggered via RIG-I or MDA-5 via MAVS than to be a real independent sensor that triggers MAVS-dependent signaling independent of the RLRs.

[Response] According to the reviewer's suggestion, we have used MDA5 KO cells and found that DDX3X rather than MDA5 was essential for circRIG-I action. In addition to scaffolding protein, recent study has demonstrated that DDX3X acts as a sensor for HIV-1 RNA (PMID: 28024153), which is consistent with our data.

The sh-Experiments that try do knock-down circRIG-I without targeting the RIG-I mRNA – to me still lack specificity (I do not believe that the shRNAs 2 and 3 here have the specificity that 1 and 4 did not have). Using pppRNA as specific RIG-I ligand as a control might help to show that the function of the RIG-I protein is really unchanged.

[Response] Actually, the aim for this experiment is to determine whether circRIG-I is essential for the increased production of type I IFN and whether the inflammatory phenotype of the *Rig-1^{fs/fs}* mutation can be explained by circRIG-I. Utilization of shRNA targeting junction site is the most classical and common strategy to knockdown circRNA. ShRNA target sequencing is thus critical for efficiency and specificity of circRNA knockdown. We found the expression of circRIG-I rather than its counterpart linear mRNA was downregulated in cell transfected with pLKO-circRIG-I-2 or pLKO-circRIG-I-3. We thus employed pLKO-circRIG-I-2 and pLKO-circRIG-I-3 to further analyze the role of circRIG-I in modulation of host antiviral immunity.

3. I would ask to make clear in figures that use human cell lines what construct are used - are they of murine or human origin. e.g. RIG-I 1-284, the overexpressed circRIG-I ect.

[Response] As suggested, we have described that we used human-derived RIG-I₁₋₂₈₄ and murine-derived circRIG-I in the Figure legend.

REVIEWER COMMENTS

Reviewer #3 (Remarks to the Author):

Overall, I think the authors have responded well to most of the comments by reviewer 1.

Regarding whether the phenotype of the RIG-I FS/FS mutation is explained by an increased amount of circRIG-I, I think it is clear that the circRNA plays a role, though it could also be influenced in other ways. But the data on DHX9 binding seems like a plausible explanation for the increased expression of the circRNA in the mutated setting.

Response Fig. 1d-g does not show that a high level of circRIG-I is highly correlated with poor outcomes of cancer patients as claimed by the authors. However, this data anyway does not seem necessary to argue against the reviewer.

I don't know why the reviewer thinks that circRIG-I is a random pick based on its chromosomal location and I think the authors argued fine against this comment.

Likewise, I think the authors argue fine against comment c.

Regarding comment d.(in which I think the reviewer must mean exon when writing exome), I find it hard to see from Response Fig. 3a if the predicted stem-loop structure that is lost due to the frameshift mutation would be expected to impact the biogenesis rate of the circRNA as the splice sites involved in backsplicing are not shown. Perhaps the authors would like to clarify this.

It is hard for me to see if fig. 4k was done on WT MEFs or in the FS/FS MEFs: I could not get this information from the main text or the figure legend. This should be clarified.

Finally, I think the authors have addressed the remaining comments about the sh-RNA experiments in a reasonable way.

Reviewer #5 (Remarks to the Author):

Regarding responses to the previous Reviewer 1 concerns, I believe they have been addressed. However, I also share the sentiment of Reviewer 1 that the mechanism is difficult to believe. The data presented suggest that the circRIG-I is an incredibly potent activator of interferon responses. In fact, in Fig 6A, circRIG-I in RIG-I fs/fs cells is proposed to facilitate the same degree of IFN- β activation as RIG-I does in VSV-infected WT cells. The functional significance of circRIG-I in IFN- β responses should be clarified in WT cells that are VSV-infected or treated with RIG-I activators such as ppp-RNA.

Point-by-point response to the reviewers' comments is as followed:

We would first like to express our sincerely thanks to the reviewers for the positive comments and affirmation of our study. We found the reviewers' comments to be very helpful and of value for improving the quality of our data and manuscript. Accordingly, we have addressed each of the comments as follows.

Reviewer's Comments:

Reviewer #3 (Remarks to the Author)

Overall, I think the authors have responded well to most of the comments by reviewer 1.

Regarding whether the phenotype of the RIG-I FS/FS mutation is explained by an increased amount of circRIG-I, I think it is clear that the circRNA plays a role, though it could also be influenced in other ways. But the data on DHX9 binding seems like a plausible explanation for the increased expression of the circRNA in the mutated setting.

Response Fig. 1d-g does not show that a high level of circRIG-I is highly correlated with poor outcomes of cancer patients as claimed by the authors. However, this data anyway does not seem necessary to argue against the reviewer.

[**Response**] We really appreciate the reviewer's professional and constructive comments. We regret any confusion caused by our inaccurate statement. As shown in **Response Figure 1a (also Figure 5b)**, we found that tumors bearing frameshift mutation of *RIG-I* expressed higher level of circRIG-I as compared with that in adjacent normal tissues. Of note, high expression level of circRIG-I was significantly associated with tumor (III/IV) stage (**Response Fig. 1c, also Figure 5d**). In addition, we also found a significant correlation between circRIG-I expression and the type of colon cancer (ulcerative colon cancer versus polypoid colon cancer) as well as the age of cancer patient (≥ 65 versus <65) (**Response Fig. 1b and Fig. 1d, also Figure 5c and Figure 5e**). Our results thus shed light on the potential role of circRIG-I in colon cancer development.

Response Figure 1. CircRIG-I is upregulated in colon cancers.

(a) RT-qPCR analysis of the relative level of circRIG-I in colon cancers and their matched adjacent normal tissues from patients with frameshift mutation of *RIG-I* (n = 3);

* $P < 0.05$, two-tailed paired Student's t-test).

(b) RT-qPCR analysis of the relative level of circRIG-I between ulcerative colon cancer (n = 44) and polypoid (n = 30) colon cancers (* $P < 0.05$, two-tailed unpaired Student's t-test).

(c) RT-qPCR analysis of the relative level of circRIG-I in colon cancers with different stage (T1/T2, n = 14; T3, n = 76; T4, n = 10; * $P < 0.05$, two-tailed unpaired Student's t-test).

(d) RT-qPCR analysis of the relative level of circRIG-I in colon cancers with different age (< 65, n = 52; ≥ 65 , n = 48; ** $P < 0.01$, two-tailed unpaired Student's t-test).

I don't know why the reviewer thinks that circRIG-I is a random pick based on its chromosomal location and I think the authors argued fine against this comment.

Likewise, I think the authors argue fine against comment c.

Regarding comment d. (in which I think the reviewer must mean exon when writing exome), I find it hard to see from Response Fig. 3a if the predicted stem-loop structure that is lost due to the frameshift mutation would be expected to impact the biogenesis rate of the circRNA as the splice sites involved in backsplicing are not shown. Perhaps the authors would like to clarify this.

[**Response**] As suggested, we have discussed the mechanism by which circRIG-I is induced by this frameshift mutation in the revised version. Although the frameshift mutation occurred on exon 3 hardly affected the sequences of back-splice/splice sites (Exon 5/Exon 12) involved in back-splicing, it impaired the stem loop structure relative to that in wildtype RNA (**Response Fig. 2a**). By analyzing published CLIP-seq data, we found that the DHX9 peaks were on exons 3-5 and exons 12-15 of *Rig-i* pre-mRNA, which were on or around the short interspersed nuclear elements (SINEs) (**Response Fig. 2b**). Considering the key role of the Alu SINEs in cis-regulation of circRNA generation, DHX9 acts as a trans-factor that suppresses RNA processing defects originating from the Alu invasion of the human genome. Our data showed that the frameshift mutation blocked DHX9-binding to the pre-RNA of *Rig-I*, thus releasing the stimulatory effects of Alu elements on back-splicing, eventually resulting in circRIG-I production (**Response Fig. 2c**).

Response Figure 2. Frameshift mutation affects the stem loop structure of pre-RNA of *Rig-I* and impairs its association with DHX9.

(a) RNA secondary structures of wild-type (WT) and frameshift mutant RIG-I pre-mRNA were predicted by RNAfold web server (<http://rna.tbi.univie.ac.at/cgi-bin/RNAWebSuite/RNAfold.cgi>). The sequences of *Rig-i* exon 3 and intron 3 were analyzed. The frameshift mutation was highlighted in red color and the pre-mRNA structural change was highlighted in blue color.

(b) Integrative Genomics Viewer (IGV) analysis of CLIP-seq coverage, DHX9 peaks at the Alu elements (SINE) of *Ddx58* were highlighted. Data range represents the coverage of uniquely mapped alignments for each profile.

(c) Interaction between DHX9 and wild-type (WT) or frameshift mutant RIG-I pre-mRNA. HEK293T cells were transfected with FLAG-DHX9 expressing vector. The biotin-labeled WT or frameshift mutation of murine *Rig-i* RNA including exon 3 and intron 3 was obtained by T7 RNA polymerase kit and Biotin RNA Labeling Mix. WT *Rig-i* RNA antisense sequence was used as control. Labeled RNA was incubated with cell lysates from HEK293T cells and RNA binding proteins were isolated by streptavidin agarose beads. The binding components were eluted by biotin, followed by SDS-Page.

It is hard for me to see if fig. 4k was done on WT MEFs or in the FS/FS MEFs: I could not get this information from the main text or the figure legend. This should be clarified.

[Response] We regret this inaccurate statement. We indeed used the WT MEFs in **Figure 4k**. And we have added this information in the legend of **Figure 4k** in the revised version.

Finally, I think the authors have addressed the remaining comments about the sh-RNA experiments in a reasonable way.

[Response] We would like to express our sincere thanks to the reviewer for the positive comments.

Reviewer #5 (Remarks to the Author)

Regarding responses to the previous Reviewer 1 concerns, I believe they have been addressed. However, I also share the sentiment of Reviewer 1 that the mechanism is difficult to believe. The data presented suggest that the circRIG-I is an incredibly potent activator of interferon responses. In fact, in Fig 6A, circRIG-I in RIG-I *fs/fs* cells is proposed to facilitate the same degree of IFN- β activation as RIG-I does in VSV-infected WT cells. The functional significance of circRIG-I in IFN- β responses should be clarified in WT cells that are VSV-infected or treated with RIG-I activators such as ppp-RNA.

[Response] We thank the reviewer for this question. We agree that circRIG-I exerts similar stimulatory effects on type I interferon response as RIG-I does. However, the results from **Figure 6A** do not indicate that circRIG-I exerts the same degree of IFN- β activation as RIG-I does. Compared with WT MEFs transfected with Scramble, *Rig-*I*^{fs/fs}* MEFs expressed higher level of IFN- β (**Response Fig. 3a**, lane 6 versus lane 5), which further confirmed that loss of RIG-I activated rather than blocked interferon response during viral infection. Of note, our data found that the expression of circRIG-I was upregulated in *Rig-*I*^{fs/fs}* MEFs. To determine whether the enhanced IFN response in *Rig-*I*^{fs/fs}* MEFs is due to circRIG-I upregulation, we thus used shRNA to specifically knockdown the endogenous circRIG-I. As expected, silence of circRIG-I significantly impaired the transduction of antiviral immune signaling in *Rig-*I*^{fs/fs}* MEFs ((**Response Fig. 3a**, lane 7 and lane 8 versus lane 6).

Response Figure 3. CircRIG-I is essential for the enhanced interferon response in *Rig-*I*^{fs/fs}* MEFs.

(a) RT-qPCR analysis of the mRNA level of *Ifnβ* in wild-type (WT) and *Rig-*I*^{fs/fs}* MEFs in presence or absence of circRIG-I (n = 3 biological replicates, mean \pm s.e.m., ****P* < 0.001, *****P* < 0.0001, two-tailed paired Student's t-test).

As the reviewer's suggestion, we have used shRNA to selectively knockdown the expression of circRIG-I in both WT and *Rig-*I*^{fs/fs}* MEFs and then treated these cells with or without VSV virus. Compared with WT MEFs, higher levels of *Ifnβ* and *Ifitm2* were

stimulated in *Rig-1^{fs/fs}* MEFs upon exposure to VSV (**Response Fig. 4a and 4b**, lane 10 versus lane 7). Moreover, silence of circRIG-I to some extent impaired type I IFN response in WT or *Rig-1^{fs/fs}* MEFs (**Response Fig. 4a and 4b**). Accordingly, increased viral titer was detected in both WT and *Rig-1^{fs/fs}* MEFs when endogenous circRIG-I was silenced (**Response Fig. 4c**). Our data thus demonstrate that circRIG-I exerts stimulatory effects on host innate immune response.

Response Figure 4. CircRIG-I exerts stimulatory effects on type-I interferon response.

(a-c) RT-qPCR analysis of the RNA levels of *Ifnβ*, *Ifitm2* as well as VSV in wild-type (WT) and *Rig-1^{fs/fs}* MEFs in presence or absence of circRIG-I under quiescent condition or viral infection (n = 4 biological replicates, mean ± s.e.m., ****P* < 0.001, *****P* < 0.0001, two-tailed paired Student’s t-test). **The Response Figure 4a and 4b** were also shown in **Figure 6a** in the revised version.

Collectively, these data supported the proposal that both RIG-I and circRIG-I exert stimulatory effects on type-I IFN response in wild-type cells upon exposure to virus, in spite of the suppressive effects of linear *RIG-I* mRNA on circRIG-I generation (**Response Fig. 5**). When endogenous circRIG-I was knockdown by shRNAs in wild-type cells, the stimulatory effects of circRIG-I on IFN-I response was attenuated, while no additional RIG-I was induced during this process, thus resulting in the reduction of IFN-I production. However, the frameshift mutation of *RIG-I* alleviated the suppression of circRIG-I by its linear counterpart and resulted in circRIG-I upregulation, eventually leading to profound IFN-I production and severe inflammatory response.

Response Figure 5. The model of circRIG-I in modulation of type I IFN response.

REVIEWERS' COMMENTS

Reviewer #5 (Remarks to the Author):

The authors have included the additional experiments requested.